# LEARNING CONTINUALLY BY SPECTRAL REGULARIZATION

**Alex Lewandowski**[1,5]     **Michał Bortkiewicz**[2]     **Saurabh Kumar**[3]

**András György**[4] **Dale Schuurmans**[1,4,5,6], **Mateusz Ostaszewski**[2] **Marlos C. Machado**[1,5,6]

[1]University of Alberta, [2]Warsaw University of Technology, [3]Stanford University,
[4]Google DeepMind, [5]Amii, [6]Canada CIFAR AI Chair

## ABSTRACT

Loss of plasticity is a phenomenon where neural networks can become more difficult to train over the course of learning. Continual learning algorithms seek to mitigate this effect by sustaining good performance while maintaining network trainability. We develop a new technique for improving continual learning inspired by the observation that the singular values of the neural network parameters at initialization are an important factor for trainability during early phases of learning. From this perspective, we derive a new *spectral regularizer* for continual learning that better sustains these beneficial initialization properties throughout training. In particular, the regularizer keeps the maximum singular value of each layer close to one. Spectral regularization directly ensures that *gradient diversity* is maintained throughout training, which promotes continual trainability, while minimally interfering with performance in a single task. We present an experimental analysis that shows how the proposed spectral regularizer can sustain trainability and performance across a range of model architectures in continual supervised and reinforcement learning settings. Spectral regularization is less sensitive to hyperparameters while demonstrating better training in individual tasks, sustaining trainability as new tasks arrive, and achieving better generalization performance.

## 1 INTRODUCTION

A longstanding goal of machine learning research is to develop algorithms that can learn *continually* and cope with unforeseen changes in the data distribution (Ring, 1994; Thrun, 1998). Current learning algorithms, however, struggle to learn from dynamically changing targets and are unable to adapt gracefully to unforeseen changes in the distribution during the learning process (Abbas et al., 2023; Dohare et al., 2024; Lyle et al., 2023; Zilly et al., 2021). Such limitations can be seen as a byproduct of assuming, one way or another, that the problem is stationary. Recently, there has been growing recognition of the fact that there are limitations to what can be learned from a fixed and unchanging dataset (Hoffmann et al., 2022), that there are implicit non-stationarities in many problems of interest (Igl et al., 2021), and that some real-world problems benefit from learning continually (Han et al., 2022; Janjua et al., 2023).

The concept of plasticity has been receiving growing attention in the continual learning literature, where the loss of plasticity—either a reduction in a neural network's ability to train (Dohare et al., 2021; Elsayed and Mahmood, 2024; Lyle et al., 2022), or ability to generalize (Ash and Adams, 2020; Zilly et al., 2021)—has been noted as a critical shortcoming in current learning algorithms. Learning algorithms, and more specifically neural networks, that are performant in the non-continual learning setting, often struggle when applied to continual learning problems. Settings where a neural network must continue to learn after changes occur in the data distribution exhibit a striking loss of plasticity such that learning slows down (Lyle et al., 2023) or even halts after successive changes (Abbas et al., 2023; Dohare et al., 2024; Nikishin et al., 2022).

Several aspects of a learning algorithm have been found to contribute to or mitigate loss of plasticity. Examples include the type of optimizer (Dohare et al., 2024; Lyle et al., 2023), the step-size (Ash and Adams, 2020; Berariu et al., 2021), the number of optimiser iterations (Lyle et al., 2023), and the use of specific regularizers (Dohare et al., 2021; Kumar et al., 2023; Lewandowski et al., 2023; Lyle et al., 2022). Such factors hint that there might be simpler underlying optimisation principles that govern the loss of plasticity. For example, the success of several methods that regularize neural

networks towards properties of the initialization suggests that some of those properties mitigate loss of plasticity (Dohare et al., 2021; Kumar et al., 2023; Lyle et al., 2022).

The properties of neural network weights at initialization are associated with trainability in the early phases of learning. Previous analyses have demonstrated that the trainability of deep neural networks can be improved by ensuring that the initialization variance of hidden activations and gradients remains uniform across the different layers of the neural network (Glorot and Bengio, 2010), which has the effect of keeping the average singular value of the layerwise Jacobians close to one. A stronger condition is dynamical isometry, where all singular values are close to one, or exactly one in the case of an orthogonal initialization (Pennington et al., 2017; Saxe et al., 2014; Xiao et al., 2018). Over the course of learning, the singular values deviate from their initialization and tend to grow over time (Martin and Mahoney, 2021), which can reduce trainability and impede continual learning.

Given the effectiveness of initialization in ensuring the trainability of a neural network, continual learning algorithms may benefit from sustaining the relevant properties of the weights. This is the motivation of several methods that use the initialization either through regularization (Kumar et al., 2023) or weight reinitialization (Dohare et al., 2021; 2024; Sokar et al., 2023). Such methods leverage the initialization explicitly, either by regularizing parameters towards their initial values or by resetting parameters by resampling them from the initialization distribution. In this paper, we seek to more directly address the loss of trainability by sustaining the key *properties* present at initialization, thereby striking a better balance between trainability and performance compared to approaches that replicate the initialization explicitly.

We investigate the key properties of the initialization that ensure trainability, how these properties are lost over the course of learning, and the effects this has on continual learning. We identify that deviations from the initial singular value distribution can result in low gradient diversity, thereby impeding continual learning. Based on this analysis, we then introduce a spectral regularizer to control the deviation of the singular values by keeping the maximum singular value of each layer close to one to directly address the deviation of the singular values from initialization over the course of learning. Our experiments show that spectral regularization is more performant and less sensitive to hyperparameters than other regularizers across datasets, nonstationarities, and architectures. While well-tuned regularizers are often able to mitigate loss of plasticity to a varying degree, learning continually with spectral regularization is robust, achieving high initial and sustained performance. In particular, we show that spectral regularization is also capable of improving generalization with both Vision Transformer (Dosovitskiy et al., 2021) and ResNet-18 (He et al., 2016) on continual versions of tiny-ImageNet (Le and Yang, 2015), CIFAR10, CIFAR100 (Krizhevsky, 2009), and SVHN2 (Netzer et al., 2011). Note that these datasets and architectures encompass *all* continual supervised learning experimental settings considered in the loss of plasticity literature. We also show that spectral regularization can improve the performance of soft-actor critic (Haarnoja et al., 2018) in reinforcement learning settings where loss of plasticity occurs due to primacy bias (Nikishin et al., 2022).

## 2 PROBLEM SETTING

We investigate the trainability of neural network learning algorithms in the task-agnostic setting. We denote a neural network with $L$ layers, defined recursively as $f_\theta(x) := h_L(x)$, where $h_0(x) = x$ is the input vector, $h_{l+1}(x) = \phi(\mathbf{W}_l h_l(x) + \mathbf{b}_l)$ with an element-wise activation function, $\phi$, and parameters $\theta = \{\theta_L, \ldots, \theta_1\}$ where $\theta_l = \{\mathbf{W}_l, \mathbf{b}_l\}$ is the weight and bias parameter for layer $l$.[1] We assume that $f_\theta$ is trained over a sequence of tasks: the $\tau$th task is specified by a distribution $p_\tau$ over the observation-target pairs, denoted by $(x, y)$. For simplicity we assume that the task (data distribution) changes periodically after every $T$ iterations. In addition, we consider the task-agnostic setting, where the learning algorithm does not have access to information about the task except through the data that it samples. For each iteration in the task, $t \in ((\tau - 1)T, \tau T]$, the learning algorithm optimises the neural network's parameters to minimize the objective

$$J_\tau(\theta) = \mathbb{E}_{(x,y) \sim p_\tau} \left[ \ell(f_\theta(x), y) \right],$$

for some loss function $\ell$. In this paper, we consider gradient-based methods to learn the weight parameters; the basic version of gradient descent would update the parameters at iteration $t$ as $\theta^{(t+1)} = \theta^{(t)} - \alpha \nabla_\theta J_{\tau(t)}(\theta)\big|_{\theta=\theta^{(t)}}$, where $\tau(t) = \lfloor t/T \rfloor$ denotes the current task number. In practice this is done by considering the empirical error instead of $J_{\tau(t)}$ using samples from the task distribution, $(x_t, y_t) \sim p_\tau$, and usually some more involved optimization algorithm.

---

[1]In this notation we suppressed the presence of biases, which can effectively be included in the parameters.

The evaluation of the continual learning algorithm is performed at the end of a task (Lyle et al., 2023). Specifically, the learning algorithm is evaluated at times $t = \tau T$, after being given $T$ iterations to learn on task $\tau$. A useful assumption in some problems is that each task is sampled independently and identically, and that each task is equally difficult, meaning that a neural network which is randomly initialized on each task is able to reach a similar objective value for any particular task (Dohare et al., 2024; Elsayed and Mahmood, 2024; Lyle et al., 2023). While this assumption is useful, it is not always applicable. In class-incremental learning (Van de Ven et al., 2022), the addition of more classes increases the difficulty of subsequent tasks. Another example is the transient tasks induced over the course of reinforcement learning (Igl et al., 2021), which can have varying task difficulty and lead to loss of plasticity due to primacy bias (Nikishin et al., 2022). We also consider such settings here.

Loss of plasticity in the continual learning literature can refer to either loss of trainability (Dohare et al., 2021; Lyle et al., 2023) or to loss of generalization (Ash and Adams, 2020). Because trainability is a requirement for learning and generalization, we focus primarily on loss of trainability. Specifically, we use loss of trainability to refer to the phenomenon that the objective value, $J_\tau(\theta^{(\tau T)})$, increases as a function of the task $\tau$. Equivalently, the performance measures, such as accuracy, decrease with new tasks. Under the assumption that the tasks are sampled independently and identically, this would suggest that the neural network's trainability diminishes on new tasks.

## 3 Spectral Properties and Continual Trainability

At a high level, the reason behind loss of plasticity is simple: the solution of one task becomes the initialization for learning on the next task, and if this initialization is not sufficiently good to enable learning in the new task, we face the aforementioned problem. To avoid this issue, we need to keep the network parameters within a region that can serve as good initialization at the start of a task. This approach comes with two challenges: (i) determining a suitable region for parameter initialization on any given task; (ii) ensuring that training within the initialization region does not prevent effective learning on the current task.

It is easy to satisfy (i) and (ii) in isolation: using a standard initialization algorithm, such as those by He et al. (2015); Hinton and Salakhutdinov (2006), and then keeping the parameters fixed at this value satisfies (i) but clearly does not satisfy (ii). Conversely, not restricting the parameter space while using a standard learning algorithm addresses (ii) but not (i). We seek a balance of the two requirements. Note that (i) and (ii) are not binary properties; hence there can be a trade-off (e.g., the parameter that allows for better optimization on the current task may be less favourable as a parameter initialization for the next task). It is worth noting that such an approach has been successfully applied in the context of online convex optimization in changing environments, where several algorithms can be recognized as running a standard learning algorithm (such as mirror descent) on a carefully selected subset of the parameter space (e.g., György and Szepesvári, 2016; Herbster and Warmuth, 1998; Zinkevich, 2003).

In this paper, to address problem (i), we first identify key properties that initialization algorithms (He et al., 2015; Hinton and Salakhutdinov, 2006) impose on the starting parameters of a neural network. Then, we propose *spectral regularization* to sustain these properties while striking a good balance between (i) and (ii).

### 3.1 Spectral properties at initialization

Neural network initialization is key to trainability (He et al., 2015; Hinton and Salakhutdinov, 2006). One property of the initialization thought to be important is that the layerwise mapping, $h_{l+1} = \text{ReLU}(\theta_l h_l)$, has a Jacobian with singular values that are close to or exactly one (Glorot and Bengio, 2010; Pennington et al., 2017; Saxe et al., 2014; Xiao et al., 2018). Writing this Jacobian explicitly, we have that $\mathbf{J}_l = \frac{\partial h_{l+1}}{\partial h_l} = \mathbf{D}_l \theta_l$ where $\mathbf{D}_l = \text{Diag}(\text{ReLU}'([\theta_l h_l]_1), \dots, \text{ReLU}'([\theta_l h_l]_d))$. [2] We can obtain upper and lower bounds on the singular values of the layerwise Jacobian in terms of the singular values of the weight matrix. Denoting the ordered singular values of $\theta_l$ and $\mathbf{D}_l$ by $\sigma_d(\theta_l) \leq \cdots \leq \sigma_1(\theta_l)$ and $\sigma_d(\mathbf{D}_l) \leq \cdots \leq \sigma_1(\mathbf{D}_l)$, respectively, we have $\sigma_d(\mathbf{D}_l)\sigma_i(\theta_l) < \sigma_i(\mathbf{J}_l) < \sigma_1(\mathbf{D}_l)\sigma_i(\theta_l)$ for all $i \in \{1, \dots, d\}$ (Zhang, 2011, Theorem 8.13). In particular, if the spectral norm (largest singular value) of the weight matrix $\theta_l$ increases, then the spectral norm of the Jacobian $\mathbf{D}_l$ increases as well, potentially impacting trainability. Furthermore, the condition number $\kappa(\mathbf{J}_l) = \sigma_1(\mathbf{J}_l)/\sigma_d(\mathbf{J}_l)$ can be bounded with the product of the condition numbers of $\theta_l$ and $\mathbf{D}_l$, $\kappa(\theta_l)$

---

[2] $\text{ReLU}'(x)$ denotes the derivative of the ReLU function, and it equals to 1 for $x > 0$ and 0 for $x < 0$ (we define it to be 1 for $x = 0$).

and $\kappa(\mathbf{D}_l)$ as $\kappa(\theta_l)/\kappa(\mathbf{D}_l) \leq \kappa(\mathbf{J}_l) \leq \kappa(\theta_l)\kappa(\mathbf{D}_l)$. Thus, if our goal is to keep the singular values of the Jacobian close to one by controlling the singular values of the weight matrix, we should ensure that the condition number of the latter is not too large.

## 3.2 AN ILLUSTRATIVE EXAMPLE

To make these points more concrete, consider a single-layer neural network $f_\theta(x) = \theta_2\texttt{ReLU}(\theta_1 x)$ mapping $x \in \mathbb{R}^2$ to $\mathbb{R}$. Suppose that the first task is to fit $x = (1,0)^\top$ and $y = 0$ in mean-squared error. An optimal solution with $f_\theta(x) = y$ is $\theta_1 = \begin{bmatrix} 1 & 0 \\ 0 & a \end{bmatrix}$ and $\theta_2 = (0, a)$ for an arbitrarily small value of $a \geq 0$. Now consider that the next task is to fit $x = (0,1)^\top$ and $y = 1$. The gradients of the loss $\ell(f_\theta(x), y) = \frac{1}{2}(y - \ell(f_\theta(x)))^2$ are, for the given new $(x, y)$ pair, $\nabla_{\theta_2}\ell(f_\theta(x), y) = (f_\theta(x) - y)\texttt{ReLU}(\theta_1 x) = (a^2 - 1)(0, a)^\top$ and $\nabla_{\theta_1}\ell(f_\theta(x), y) = (f_\theta(x) - y)x\texttt{ReLU}'(\theta_1 x)^\top = (a^2 - 1)\begin{bmatrix} 0 & 0 \\ 0 & a \end{bmatrix}$. Performing updates with these gradients keeps the $\Theta(a)$ parameters at the same order while maintaining a loss of approximately 1. The condition number of the weight matrix $\theta_1$ is $\kappa = 1/a$, which requires $\Theta(1/a) = \Theta(\kappa)$ update steps to make significant progress during learning. (Another example showing this empirically is given in Appendix A.1.)

## 3.3 TRAINABILITY AND EFFECTIVE GRADIENT DIVERSITY

We can generalize the above observation to a problem of reduced *gradient diversity*. By gradient diversity, we mean the spread of singular vectors in the matrix of per-example stochastic gradients, $\mathbf{G} = [\mathbf{g}_1, \ldots, \mathbf{g}_m]$. If this matrix contains a few large singular values and many small singular values, then the gradients will be largely in the span of the singular vectors corresponding to the large singular values. For this analysis, we focus on the rank, which is one particular summary statistic for the set of singular values that counts the number of non-zero singular values, $\text{rank}(\mathbf{G}) = |\{i : \sigma_i(\mathbf{G}) > 0\}|$. However, for practical purposes, the rank is problematic because it is unstable to perturbations (Feng et al., 2022, Theorem 1). In our experiments below we will consider the condition number, $\sigma_1(\mathbf{G})/\sigma_m(\mathbf{G})$ and effective rank, $\text{erank}(\mathbf{G}) = \sum \bar{\sigma}_i(\mathbf{G}) \log \bar{\sigma}_i(\mathbf{G})$, where $\bar{\sigma}_i(\mathbf{G}) = \sigma_i(\mathbf{G})/\sum_i \sigma_i(\mathbf{G})$ (Roy and Vetterli, 2007). These measures make explicit the issues that arise if the largest singular value grows faster than the smallest singular values (our experiments show this to be the case in Section 5.2). In this case, the condition number increases and the effective rank decreases. We refer to this as a reduction in the *effective gradient diversity*.

A reduction in the rank, or erank, of the gradient matrix results in collinear gradients on different datapoints, limiting the diversity of gradient directions used in the parameter update. This can have an adverse effect on learning. In the extreme case, where the gradient matrix is rank one, every datapoint provides a gradient in the same direction, even if the datapoints correspond to different classes. Consider the gradient of the loss on a particular datapoint $(x_i, y_i)$ with respect to the weight matrix of a hidden layer, $\theta_l \in \mathbb{R}^{d \times d}$, which can be written recursively for the parameters of layer $l$ as $\mathbf{G}_{l,i} = \nabla_{\theta_l}\ell(f_\theta(x_i), y_i) = \delta_{l,i}h_{l-1,i}^\top$, where $\delta_{l,i} = \theta_{l+1}^\top\delta_{l+1,i}\mathbf{D}_{l,i}$ is the error gradient from the next layer with $\delta_{L,i} = \partial\ell(f_\theta(x_i), y_i)/\partial f_\theta$ and $\mathbf{D}_{l,i} = \text{Diag}(\phi'([\theta_l h_{l,i}]_1), \ldots, \phi'([\theta_l h_{l,i}]_d))$. We can rewrite the gradient in terms of its vectorization $\mathbf{g}_{l,i} = \text{vec}(\mathbf{G}_{l,i}) = (\mathbf{I}_d \otimes \theta_{l+1}^\top)\mathbf{v}_{l,i}$ where $\mathbf{I}_d$ is the $d \times d$ identity matrix and $\mathbf{v}_{1,i} = \text{vec}(\delta_{l+1,i}\mathbf{D}_{l,i}h_{l-1,i}^\top) \in \mathbb{R}^{d^2}$ is the vectorization of data-dependent terms. The matrix of gradients for different datapoints is the concatenation of the per-example gradients, $\mathbf{G}_l = [\mathbf{g}_{l,1}, \mathbf{g}_{l,2}, \ldots, \mathbf{g}_{l,m}] = (\mathbf{I}_d \otimes \theta_{l+1}^\top)\mathbf{V}_l$ where $\mathbf{V}_l = [\mathbf{v}_{l,1}, \mathbf{v}_{l,2}, \ldots, \mathbf{v}_{l,m}]$. The rank of $\mathbf{G}_l$ is upper bounded as

$$\text{rank}(\mathbf{G}_l) < \min\{\text{rank}(\mathbf{I}_d \otimes \theta_{l+1}^\top), \text{rank}(\mathbf{V}_l)\} = \min\{d\,\text{rank}(\theta_{l+1}^\top), \text{rank}(\mathbf{V}_l)\}.$$

Thus, if the (effective) rank of $\theta_{l+1}$ decreases, then the (effective) rank of the gradient matrix may decrease as well. The rank of the gradient matrix can even decrease due to rank reduction in parameters at other layers, or through the rank of the representation, which others have noted to occasionally correlate with loss of trainability (Dohare et al., 2024; Kumar et al., 2023; Lyle et al., 2023).

**Why Do Spectral Properties Deviate From Initialization?** In large-scale self-supervised learning, it has been observed empirically that the parameter norm grows at a rate of $\sqrt{t}$, where $t$ is the number of iterations (Merrill et al., 2021). Similar observations have been made in continual learning, provided that the neural network does not stop learning due to loss of trainability (Dohare et al., 2024; Lyle et al., 2024; Nikishin et al., 2022). A growing parameter norm is problematic from an

optimization perspective. Specifically, the parameter norm that these works consider is the Frobenius norm, which is equal to the sum of squared singular values, $\|\theta\|_F^2 = \sum_i \sum_j [\theta]_{ij}^2 = \sum_i \sigma_i(\theta)^2$. Thus, a growing parameter norm is equivalent to an increasing sum of the squared singular values. In particular, note that $\|\theta_l\|_F \leq \sqrt{\operatorname{rank}(\theta_l)}\sigma_1(\theta_l)$ where $\sigma_1(\theta_l)$ denotes the largest singular value, or the spectral norm (Golub and Van Loan, 2013). If the parameter norm of layer $l$ grows at a rate of $\sqrt{t}$, then the spectral norm of the parameter matrix $\theta_l$ also increases at the same rate. This leads to an increase in the spectral norm of the layerwise Jacobian, $\sigma_d(\mathbf{D}_l)\sigma_1(\theta_l) \leq \sigma_1(\mathbf{J}_l)$, which can reduce effective gradient diversity and may harm trainability.

## 4 SPECTRAL REGULARIZATION FOR CONTINUAL LEARNING

If important properties of the initialization are lost during the course of learning, it is natural to regularize the neural network toward the initialization. This is the motivation for regularization (Kumar et al., 2023) and weight reinitialization (Dohare et al., 2024; Sokar et al., 2023) as loss of plasticity mitigators. However, our motivation is to more directly target key properties of initialization, using the insights from Section 3.

We denote a regularizer by $\mathcal{R}_{\tau(t)}(\theta, s)$, which is a function of (i) the parameters of the neural network, $\theta$, (ii) the data, through the current task $\tau(t)$, and of (iii) auxiliary information, such as the parameters at initialization, through the state variable $s$. Naturally, not every regularizer takes all of these elements into account. The regularizer is optimized alongside the base objective, $J_{\tau(t)}(\theta)$. We write the composite objective as $J_{\tau(t)}^\lambda(\theta) = J_{\tau(t)}(\theta) + \lambda\mathcal{R}_{\tau(t)}(\theta, s)$, where $\lambda$ is a tunable hyperparameter governing the regularization strength.

In Section 3, we argued that a growing spectral norm and condition number can harm trainability by reducing effective gradient diversity and that this growth occurs at a rate of $\sqrt{t}$. Now we investigate regularization as a means of controlling the spectral norm to maintain the trainability of the neural network.

One commonly used regularizer in both continual and non-continual learning is L2 regularization, $\mathcal{R}_{\tau(t)}(\theta, \varnothing) = \|\theta\|^2$, which regularizes the parameters towards zero. A recent alternative, L2 regularization towards the initialization, $\mathcal{R}_{\tau(t)}(\theta, \theta^{(0)}) = \|\theta - \theta^{(0)}\|^2$, was proposed to deal with sensitivity to parameters near zero (Kumar et al., 2023). Regularizing towards the particular parameters present at initialization allows the neural network to regenerate parameters, providing a soft reset to parameters if the gradient on the base objective is zero (see Appendix A.2 for an example and Appendix A.4 for more details). One potential problem with L2 regularization towards the initialization is that it may prevent the parameters from deviating from the particular sampled value from the initialization distribution.

Our proposed spectral regularizer explicitly regularizes each layer's spectral norm (the maximum singular value) and addresses the reduced effective gradient diversity described in Section 3.3. We only minimize the maximum singular value because the smallest singular value remains relatively constant across all the architectures we consider (see Appendix C.6). The parameter matrix $\theta_l$ in layer $l$ is a concatenation of the weight matrix $\mathbf{W}_l$ and the bias $\mathbf{b}_l$, giving the augmented parameter matrix $\theta_l = [\mathbf{W}_l|\mathbf{b}_l]$ and augmented input $\tilde{h}_l = [h_l, \mathbf{1}]$. With this notation, the equation defining the neural network $f_\theta$ can be rewritten as $h_{l+1}(x) = \phi(\mathbf{W}_l h_l(x) + \mathbf{b}_l) = \phi(\theta_l \tilde{h}_l)$. As discussed in Section 3, our goal is to control the largest singular value of $\theta_l$. Using the fact that the concatenated parameters can be upper-bounded by the sum, $\sigma_1(\theta_l) = \sigma_1([\mathbf{W}_l|\mathbf{b}_l]) \leq \sigma_1(\mathbf{W}_l) + \sigma_1(\mathbf{b}_l)$, we achieve this by regularizing the spectral norm of each parameter in the layer separately: the spectral norm of the multiplicative weight parameter $\mathbf{W}_l$ is regularized towards one and the spectral norm of the additive bias parameter $\mathbf{b}_l$ is regularized towards zero:

$$\mathcal{R}_{\tau(t)}(\theta_l) = \sum_{l \in \text{layers}} \left( (\sigma_1(\mathbf{W}_l)^k - 1)^2 + \left(\sigma_1(\mathbf{b}_l)^k\right)^2 \right) = \sum_{l \in \text{layers}} \left( (\sigma_1(\mathbf{W}_l)^k - 1)^2 + \|\mathbf{b}_l\|_2^{2k} \right). \quad (1)$$

We add the exponent $k$ to the spectral norm to penalize large spectral norms deviating from one. We set $k = 2$ in our experiments and provide an ablation study in Appendix C.1. For implementation details for other layers, such as normalization and convolutional layers, see Appendix A.7. The largest singular value can be computed efficiently using power iteration (Golub and Van der Vorst, 2000; Householder, 2013) with a computational complexity scaling linearly in the number of parameters.

[3] Similar to previous work using spectral regularization for generalization in supervised learning, we find that a single iteration is sufficient for effective regularization (Yoshida and Miyato, 2017).

In addition to the exponent $k$, our approach to spectral regularization is different in two important ways compared to previous work, such as spectral-norm regularization (Yoshida and Miyato, 2017). First, we regularize the spectral norm of every parameter, including bias terms and normalization parameters (see Appendix A.7). This is required because every parameter experiences norm growth. Second, we regularize the spectral norm of the multiplicative parameters to one rather than zero. Aggressive regularization of the spectral norm towards zero can lead to collapse issues similar to L2 regularization towards zero. These differences are crucial to improving trainability in continual learning. We emphasize that regularizing the maximum singular value is sufficient for sustaining trainability, and that other approaches such as L2 can lead to over regularization and increased sensitivity. Lastly, spectral regularization is preferable over spectral normalization (Miyato et al., 2018; Zhai et al., 2023), which explicitly normalizes the weights in the forward pass to be exactly one, for two reasons: (i) spectral normalization is data-dependent (see Section 2.1 and Equation 12 by Miyato et al. (2018), and Appendix A.6), which can be problematic in continual learning due to the changing data distribution, and, (ii) other forms of normalization are already used to train deep neural networks, such as LayerNorm (Ba et al., 2016), and it has been shown that spectral normalization does not improve trainability (Lyle et al., 2023). See Appendix A.5 for detailed comparison to L2 regularization and spectral normalization.

## 5 EXPERIMENTS

The goal of our experiments is to investigate the effect of spectral regularization on trainability in continual supervised learning, as well as reinforcement learning. We cover several different datasets, types of nonstationarity, and architectures. We compare our proposed regularizers against baselines that have been shown to improve trainability in previous work, which we detail below. Overall, our experiments demonstrate that spectral regularization (i) consistently mitigates loss of trainability on a wide variety of continual supervised learning problems, including training large neural networks for thousands of epochs across a hundred tasks, (ii) is highly robust to the regularization strength, type of non-stationarity and the number of training epochs per task, (iii) achieves better generalization performance over the course of continual learning, and (iv) is generally applicable, which we demonstrate by applying spectral regularization to reinforcement learning with continuous actions.

**Datasets, Nonstationarities, and Architectures** Our main results uses *all* commonly used image classification datasets for continual supervised learning: tiny-ImageNet (Le and Yang, 2015), CIFAR10, and CIFAR100 (Krizhevsky, 2009). Experiments in the appendix also use smaller-scale datasets, like MNIST (LeCun et al., 1998), Fashion MNIST (Xiao et al., 2017), EMNIST (Cohen et al., 2017), and SVHN2 (Netzer et al., 2011). In addition to the dataset, we consider different types of non-stationarity: (i) random label assignments, (ii) pixel permutation, (iii) label flipping, and (iv) class-incremental learning. Random label assignments are commonly used to evaluate trainability (Lyle et al., 2023) due to the large distribution shift between tasks in memorizing completely new and random labels. Pixel permutations, on the other hand, require only learning the permutation mask applied to the image; it induces loss of trainability more slowly but is useful for evaluating generalization (Kumar et al., 2023). Label flipping is a re-assignment of all the observations from one label to another label. Unlike the other two non-stationarities, a label flip distribution shift does not require learning a new representation. This is because only the output layer of a neural network needs to be permuted to learn the label re-assignment, but gradients can still unnecessarily change the representation leading to loss of trainability (Elsayed and Mahmood, 2024). We also consider the class-incremental setting in which the network is trained on a growing subset of the classes from a dataset, starting with only five classes on the first task and introducing five new classes on new tasks. We use both a ResNet-18 (He et al., 2016) and Vision Transformer (Dosovitskiy et al., 2021).

**Loss of Trainability Mitigators** In our main results, we compare spectral regularization against L2 regularization towards zero, shrink and perturb (Ash and Adams, 2020), L2 regularization towards the initialization (Kumar et al., 2023), recycling dormant neurons (ReDO, Sokar et al., 2023), concatenated ReLU (Abbas et al., 2023; Shang et al., 2016), and Wasserstein regularization (Lewandowski et al., 2023). Several regularizers in the continual learning without forgetting literature rely on privileged task information, which is not applicable to the task-agnostic setting that we consider. We use the streaming conversion (Elsayed and Mahmood, 2024) to transform elastic weight

---

[3]On a 1080TI, training with spectral regularization is approximately 14% slower.

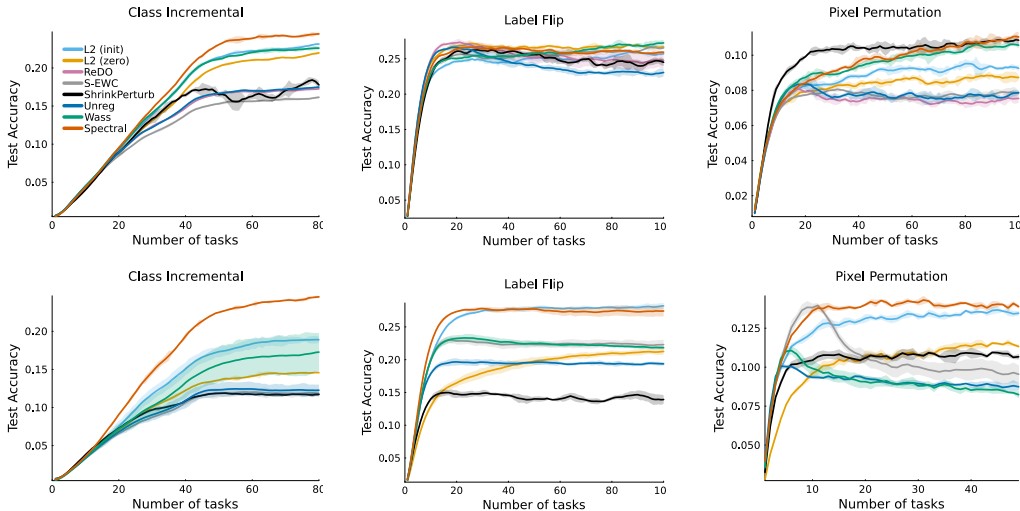

Figure 1: **Generalization across different types of non-stationarity on tiny-ImageNet using a ResNet (top) or a Vision Transformer (bottom).** Compared to the baselines, spectral regularization is consistently among the best-performing methods across class incremental, label flip, and pixel permutation non-stationarities. Note that the Vision Transformer often achieves better generalization performance than the ResNet architecture.

consolidation (Kirkpatrick et al., 2017; Zenke et al., 2017), so that it no longer requires task boundary information, and include it as a baseline. Additional experiment details can be found in Appendix B.

**Overview of Results**    Our results demonstrate that our method is (i) performant on average, and better than the best baseline in 10 out of 12 problems considered; (ii) robust across the problems considered, given that it is always in the top 2 best-performing methods (the best baseline is in the top 2 only 7 out of 12 problems). Furthermore, we also show that our method is at least on par with the best baseline on a reinforcement learning problem, where the best baseline is only better than our method at the very end of the training for the original problem and becomes superior once the problem is varied.

### 5.1    COMPARATIVE EVALUATION

In Figure 1, we plot the results of training a ResNet-18 and a Vision Transformer on tiny-ImageNet with different non-stationarities. Across networks, non-stationarities, and methods considered, we see that spectral regularization is among the methods best capable of sustaining plasticity. The advantage of our approach, spectral regularization, is particularly high in class-incremental learning, but the performance of spectral regularization with a Vision Transformer was also the best on label flipping and pixel permutation. In contrast, the performance of other baselines was highly variable with respect to the specific type of non-stationarity. For example, shrink and perturb is not robust to the settings considered. Sometimes, it is among the best-performing or worst-performing methods. Similar results on other datasets and the pixel permutation non-stationarity can be seen in Figure 2.

### 5.2    LOOKING INSIDE THE NETWORK

We now explore how the structural properties of a neural network evolve over the course of continual learning, how these properties are affected by spectral regularization, and the different baselines considered. For this, we consider the continual learning problem in which a ResNet-18 must memorize a set of random labels which changes from task to task using the tiny-ImageNet, CIFAR10, and CIFAR100 datasets. We consider the average representation change to measure the distance between the neural network's hidden activations from the beginning of one task, to the beginning of the next task. The average representation change is a proxy for plasticity, allowing us to see how much the behavior of the neural network has changed. In Figure 3, we see that the unregularized `ReLU` networks suffers from loss of trainability in all problems considered, and this coincides with an

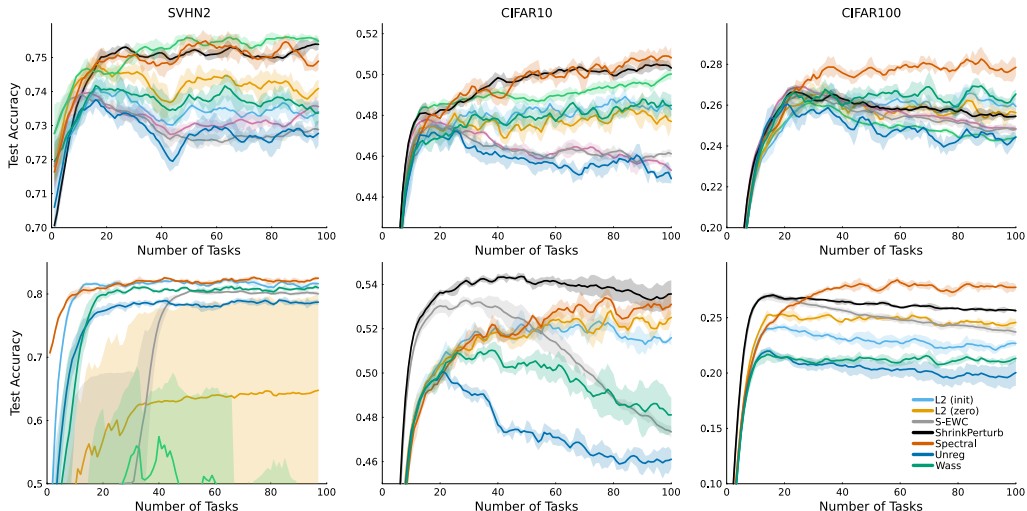

Figure 2: **Continual learning with pixel permutation tasks on SVHN2, CIFAR10, CIFAR100 using a ResNet-18 (top) or a Vision Transformer (bottom)**. Across different datasets, spectral regularization is effective at maintaining test accuracy on new tasks. Without any mitigators, both ResNet-18 and Vision Transformer have diminishing test accuracy, suggesting loss of plasticity.

increasing spectral norm (middle-top) and a decrease in the average representation change (bottom).[4] Although only spectral regularization directly regularizes the spectral norm of the network, other regularizers do so indirectly by controlling other norms, like L2. However, these other regularizers also regularize other parameters, preventing them from deviating from initialization and potentially leading to suboptimal use of capacity, which can be observed in the bottom row in Figure 3. In Appendix C.3, we show that the Vision Transformer was not able to memorize random labels, suggesting that its capabilities for generalization are offset by its relatively lower trainabilty.

## 5.3 SENSITIVITY ANALYSIS

The effectiveness of a regularizer depends on the regularization strength. Too much regularization can slow down training, leading to suboptimal performance on any given task. However, too little regularization can lead to loss of trainability and suboptimal performance on future tasks. In a continual

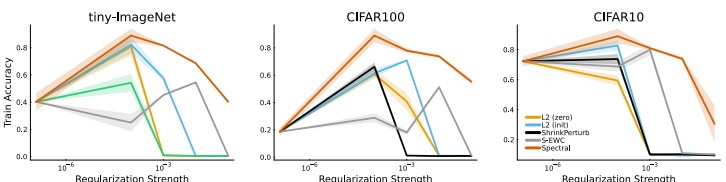

Figure 4: **Sensitivity analysis on regularization strength.** Compared to other regularizers, spectral regularization is insensitive to regularization strength while sustaining higher trainability for any given regularization strength.

learning problem, hyperparameter tuning is particularly costly (or infeasible), especially if the distribution over tasks is not known in advance (Mesbahi et al., 2024). Thus, we should favour regularizers that are less sensitive to hyperparameters. Our results in Figure 4 show that the sensitivity of spectral regularization to its hyperparameter is much lower than other regularizers. In Appendix C.4, we also present additional results showing the robustness of spectral regularization when varying (i) the regularization strength using other network architectures, (ii) the type of non-stationarity, and (iii) the number of training epochs per task.

---

[4]In Appendix C.10, we provide a table showing the final performance after training on the sequence of tasks.

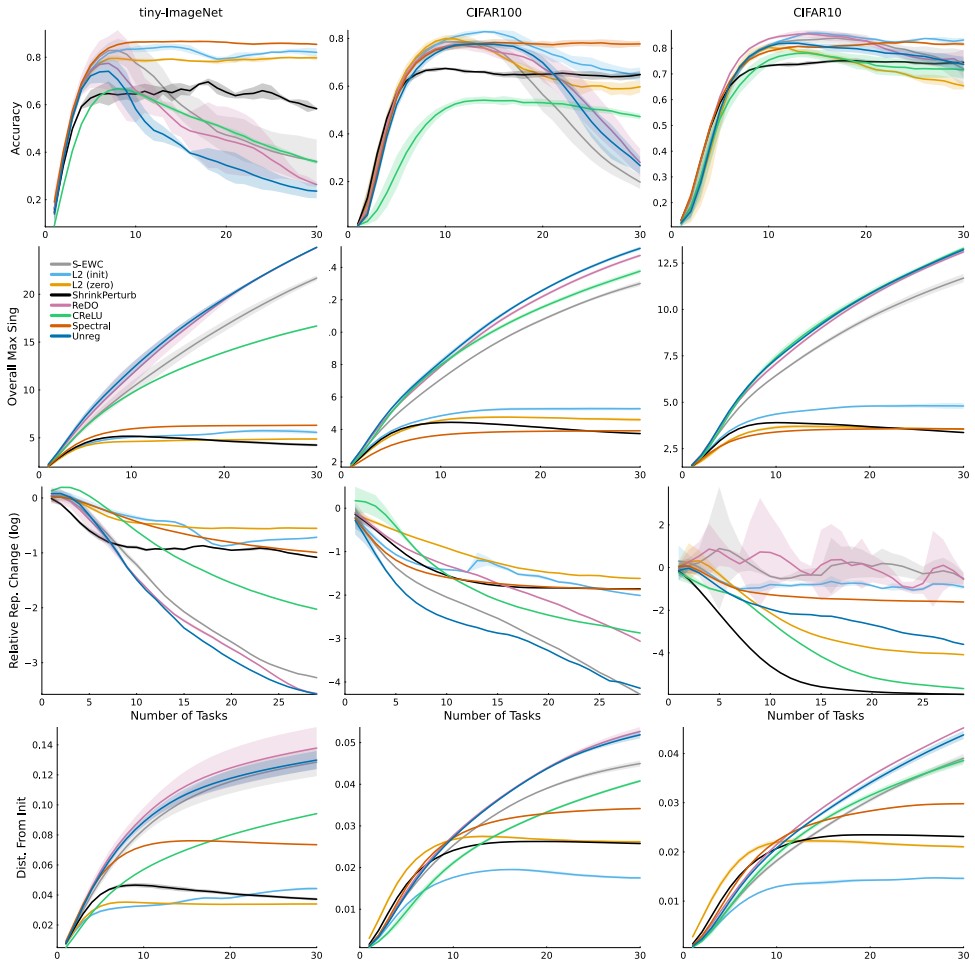

Figure 3:  **Trainability and neural network properties across ImageNet, CIFAR10, and CIFAR100.** Baselines that suffer from a loss of trainability (*top*) also have an increasing average spectral norm (*middle-top*), and a decrease in their average representation change (*middle-bottom*).

## 5.4    FROM SUPERVISED LEARNING TO REINFORCEMENT LEARNING

In addition to supervised learning, we evaluate spectral regularization in reinforcement learning. Specifically, we investigate the control tasks from the DMC benchmark (Tassa et al., 2020), with soft actor-critic  (SAC, Haarnoja et al., 2018), in the high replay-ratio (RR) regime (D'Oro et al., 2022) with 16 gradient updates for every environment step. The high replay-ratio regime leads to primacy bias (Nikishin et al., 2022), a phenomenon related to loss of plasticity. Recent work has demonstrated that only full network resets with layer normalization (Ba et al., 2016) serve as an effective mitigation strategy in this setting (Ball et al., 2023; Nauman et al., 2024).

As shown in Figure 5, spectral regularization yields substantially higher returns compared to the baseline that uses layer normalization with either an unbounded replay buffer size or a limited replay buffer size. In aggregate, spectral regularization is competitive with the final performance of the strongest baseline, full network resets in combination with layer normalization. However, in five out of seven environments, spectral regularization substantially improves over the reset baseline in terms of sample efficiency (see Appendix C.8). Spectral regularization is also less sensitive to its regularization strength than resets are for their reset frequency (Figure 5, middle-top and middle-bottom). Moreover, we used a single regularization strength for every network. Better performance may be achieved by individually tuning each regularization strength for the value and policy networks. In addition, a combination of spectral normalization and layer norm aggressively reduces Q-value overestimation (Figure 5, top-left), which is one of the well-established proxies for RL

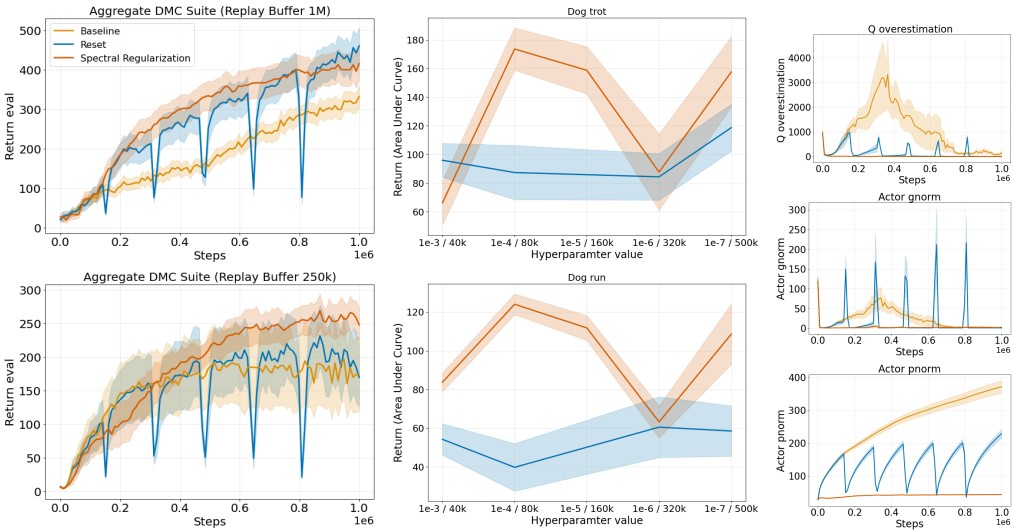

Figure 5: **Spectral regularization enhances plasticity in reinforcement learning in the DMC suite.**
Spectral regularization is competitive with the network reset + layernorm (Reset) even when the
replay buffer is unbounded (*Top Left*). When the replay buffer size is bounded to 250k steps, spectral
regularization improves over the Reset baseline (*Bottom Left*). In both cases, Spectral regularization
significantly outperforms layernorm (Baseline), Compared to the hyperparameter governing the reset
frequency, spectral regularization is less sensitive to its regularization strength. Spectral regularization
also prevents both parameter and gradients from exploding, and reduces value overestimation (*Right*).

training destabilization (Hasselt, 2010; Nauman et al., 2024). Lastly, spectral normalization prevents
exploding gradients and keeps weights' norms small during training, which is a key component for
learning continually and preserving the plasticity of neural networks (Lyle et al., 2022; 2023).

## 6 CONCLUSION

In this paper, we investigated the connection between initialization, trainability, and regularization. We
identified that deviations of the maximum singular value of each layer can lead to low gradient diver-
sity, preventing neural networks from training on new tasks. To directly preserve trainability properties
present at initialization, we proposed spectral regularization as a way of controlling the maximum
singular value of each layer so that it does not deviate significantly from 1. Our experiments show that
spectral regularization is more performant and less sensitive to hyperparameters than other methods
across datasets, nonstationarities, and architectures. While several methods mitigate loss of trainabil-
ity to a varying degree, learning continually with spectral regularization is robust, achieving high
initial and sustained performance. We also showed that spectral regularization is capable of improving
generalization with both Vision Transformer and ResNet-18 on continual versions of tiny-ImageNet,
CIFAR10, CIFAR100, amongst others. Lastly, we showed that spectral regularization learned a more
effective policy in reinforcement learning by avoiding early loss of plasticity due to primacy bias.

### ACKNOWLEDGMENTS

The research is supported in part by the Natural Sciences and Engineering Research Council of
Canada (NSERC), the Canada CIFAR AI Chair Program, Alberta Innovates, and the Digital Research
Alliance of Canada. Michał Bortkiewicz and Mateusz Ostaszewski were partially supported by the
National Science Centre, Poland, under a grant 2023/51/D/ST6/01609.

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

# A ADDITIONAL DETAILS

## A.1 ILLUSTRATIVE EXAMPLE: LARGE SPECTRAL NORM CAN IMPEDE TRAINABILITY

We show that, given a family of solutions on the first task, preference should be given to solutions that have a spectral norm that is closer to one. We consider 2 dimensional binary and orthogonal inputs, $x_1 = [0,1]^\top$ and $x_2 = [1,0]^\top$, with corresponding binary targets, $y_1 = y_2 = 1$. We consider the squared loss, and use a mutli-layer perceptron with one hidden layer and two units, $f_{w_2,w_1}(x) = w_2\text{ReLU}(w_1x)$. A common observation regarding solutions found for deep neural networks is that they are low rank (Arora et al., 2019; Galanti et al., 2022; Jacot, 2023; Timor et al., 2023). In addition, it has been observed that large outliers in the spectrum of a deep network emerge during training, meaning that the largest singular values of the weight matrices tend to be much larger than their smallest singular values (Martin and Mahoney, 2021; Pennington et al., 2018). Consider one such solution where $w_1 = \begin{bmatrix} -1 & c \\ 1/c & -1 \end{bmatrix}$, and $w_2 = [1/c, c] \in \mathbb{R}^{1\times2}$. The input-output function is invariant under different values of $c$, achieving zero error for any particular choice. However, the parameter matrix for the hidden activations, $w_1$, is low rank because it only has a single non-zero singular value that is not invariant. That is, the largest singular value of $w_1$ depends on the choice of $c$, $\|w_1\|_2 := \sigma_{max}(w_1) = c + 1/c$. In addition, the gradients and parameter updates depend on $c$. Thus, if the targets were to change, $y_1' = 0$, the imbalance of singular values due to very large or small values of $c$ can impede trainability:

$$w_1'(x_1) = \begin{bmatrix} -1 & c \\ 1/c & -1 \end{bmatrix} - \alpha \left( \underbrace{\begin{bmatrix} 0 & 1/c \\ 0 & 0 \end{bmatrix}}_{\nabla_{w_1}\ell_1} \right), \qquad w_2'(x_1) = [1/c, c] - \alpha \left( \underbrace{[c, 0]}_{\nabla_{w_2}\ell_1} \right), \qquad (2)$$

where $\ell_1 = \ell(f_{w_2,w_1}(x_1), y_1')$, and $\alpha$ is a step-size. One of the consequences of the imbalanced singular values is that the per-example gradients are poorly conditioned (Wu et al., 2021). That is, when we sample $x_1$, there will be a large gradient for $w_2$, $\|\nabla_{w_2}\ell(f_{w_2,w_1}(x_1), y_1')\|^2 = c$ but a small gradient for $w_1$, $\|\nabla_{w_1}\ell(f_{w_2,w_1}(x_1), y_1')\|^2 = 1/c$. Thus, a sufficiently small step-size that stabilizes the update to $w_2$ leads to slow learning on $w_1$. A similar result holds if the target $y_2$ were to change In Figure 6, we demonstrate that learning is more stable with a smaller spectral norm in this example. For higher dimensional weight matrices, we may expect that the imbalance of the singular values leads to low gradient diversity where particular per-example gradients give rise to outliers with a large gradient magnitudes. Thus, all else being equal, a solution with a smaller spectral norm ($c \approx 1$) is preferable for the purposes of continual learning.

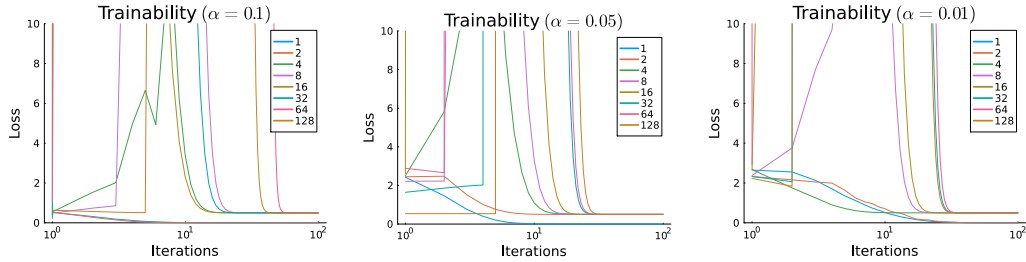

Figure 6: **Illustrative Example.** A lower spectral norm is more trainable across different step sizes in the simple illustrative example.

## A.2 DIFFERENCES BETWEEN CONTINUAL AND NON-CONTINUAL LEARNING

One critical difference between continual and non-continual learning is that, in continual learning, convergence towards a fixed point is not the objective. Due to continual changes in the data distribution, there is no fixed point that is optimal for every distribution in general. Thus, convergence to a fixed point is necessarily suboptimal. If convergence to a fixed point does occur, then the norm of the gradients will be zero. The gradients converging to zero is a condition for loss of plasticity,

and can be satisfied on every distribution depending on the choice of activation function. Focusing on the case when the activation function is $\texttt{ReLU}(x) = \max(x, 0)$, the gradients can become zero if 1) the weights converge to zero or, 2) the activations converge to zero. As a running example, we will consider the two-layer neural network without biases denoted by $f_\theta(x) = \theta_2 \phi(\theta_1 x)$, where $\theta_1 \in \mathbb{R}^{d \times d_{in}}$ and $\theta_2 \in \mathbb{R}^{d_{out} \times d}$ are weight matrices, and $\phi$ is an elementwise activation function. We will consider the squared loss, $J(\theta) = \frac{1}{2}\mathbb{E}_{(x,y) \sim p}\left[(y - f_\theta(x))^2\right]$, but a similar result holds for classification losses, such as cross-entropy. The gradient for each of the parameters is given by, $\nabla_{\theta_2} J(\theta) = (y - f)\phi(\theta_1 x)^\intercal$ and $\nabla_{\theta_1} J(\theta) = \theta_2^\intercal(f - y)\phi'(\theta_1 x)x^\intercal$. The neural network will reach a fixed point in continual learning, meaning that the gradients are zero for every datapoint, if either 1) both $\theta_1$ and $\theta_2$ are zero, or 2) the activations are zero for every input $x$, $\phi(\theta_1 x) = 0$,

### A.3 SPECTRAL NORM AND LIPSCHITZ CONSTANT

One other potential reason for why a growing spectral norm is problematic is from the theory convex optimisation. For the $\texttt{ReLU}$ activation function, the Lipschitz constant of the layer map, $h_{l+1} = \phi(\theta_l h_l)$, is equal to the spectral norm of the weight matrix. We can thus bound the Lipschitz constant of the entire neural network above by the product of the layer-wise Lipschitz constants (Szegedy et al., 2013). Putting these results together, we conclude that during continual learning, the Lipschitz constant of each layer is increasing and thus the Lipschitz constant of the entire network is also increasing. This is problematic from the standpoint of optimisation because gradient descent only converges on convex optimisation problems when the step-size is smaller than $\frac{2}{L}$, where $L$ is the Lipschitz constant. Thus, if the Lipschitz constant grows then, eventually, the chosen step-size will be too large and gradient descent will not converge locally on the task.

### A.4 MORE DETAILS ON REGULARIZATION AND TRAINABILITY

One other important advantage of regularization is that it has well understood effects for ensuring trainability in approaches outside of deep learning, such as degenerate linear regression and inverse problems (Benning and Burger, 2018).

Each regularizer biases the parameter dynamics in their own way, such as keeping the parameters close to zero or close to initialization. But, the regularizer term is independent of the base objective, and potentially the data and/or architecture. Even if the base objective provides a zero gradient, explicit regularization can provide a gradient to all weights, and reset weights if the hidden unit associated with the weight becomes saturated. For example, if the nonlinearity is $\texttt{ReLU}$, then a hidden unit $(i)$ is inactive if for all inputs in the dataset $(k)$, $h_i^l = \phi\left(\sum_{j=1}^{d_{l-1}} \theta_{ij}^l h_{j,k}^{l-1}\right) = 0$. Then the hidden representation has collapsed for that unit, and none of the weights contributing to that unit will be updated, keeping the unit inactive. But, with regularization, the weights will be updated and the unit may become active again for some inputs.

### A.5 DETAILS REGARDING SPECTRAL REGULARIZATION

**Spectral Regularization vs L2 Regularization** L2 regularization is not as effective as spectral regularization because it constrains the magnitude of the parameters, and can even cause rank collapse (Kumar et al., 2023). Spectral regularization constrains only the spectral norm of the weight matrix close to 1, $\sigma^{(1)}(\theta_l) \approx 1$. This has the effect of controlling the magnitude of the maximal column of the weight matrix. That is, if $\theta \in \mathbb{R}^{d_l \times d_{l-1}}$, then $1/\sqrt{d_l}\|\theta_l\|_1 < \|\theta_l\|_2$ where $\|\theta_l\|_1 = \max_{1 \leq j \leq d_l} \sum_{i=1}^{d_l} |[\theta_l]_{ij}|$. This implies that $\sum_{i=1}^{d_l} |[\theta_l]_{ij}| < \sqrt{d_l}$ for every column $j$. This differs in two ways from L2 regularization: (i) the regularizer does not regularize all the columns of the weight matrix, only the parameters in the column with a maximal sum, and (ii) the regularizer does not regularize the individual weights of the column, but their sum. This means that the parameters can potentially be larger, and move further from the initialization, potentially enabling more effective use of the neural network's capacity. Lastly, we note that L2 regularization can also be viewed in terms of a matrix norm, specifically the Frobenius norm.

**Spectral Regularization vs Spectral Normalization:** The spectral norm can also be controlled via spectral normalization (Miyato et al., 2018). For continual learning, spectral regularization is preferable over spectral normalization for two reasons: (i) spectral normalization is data-dependent

(see Section 2.1 and Equation 12 in Miyato et al. (2018), and Appendix A.6 for more details), which can be problematic in continual learning due to the changing data distribution, and, (ii) other forms of normalization are often already used to train deep neural networks, such as LayerNorm (Ba et al., 2016). LayerNorm, in particular, regularizes the magnitude of the layerwise map by dividing by the norm of the features in the layer. However, because the spectral norm of the underlying parameters still grows at a rate of $\sqrt{t}$, the normalization layer itself will suffer from loss of gradient diversity. For the purposes of ensuring continual trainability, regularization explicitly controls the spectral norm of all parameters and is thus preferable over normalization in isolation. That being said, normalization can still improve performance and optimal performance may be achieved by a combination of regularization and normalization, which we show in Section 5.3.

## A.6 Categorizing Regularizers for Continual Learning

We can categorize regularizers broadly into data-dependent and data-indepedent regularizers. Data-dependent regularizers use the data in some way, which can be probelmatic for continual learning when the data distribution changes. Regularizing on one distribution of data does not necessarily maintain useful properties for learning on the next distribution of data. Examples of data-dependent regularizers include feature rank regularization (Kumar et al., 2021), auxiliary tasks and other feature-space regularizers (Lyle et al., 2022). Other non-standard approaches that have a data-dependent regularization effect include gradient regularizers such as gradient-clipping and weight reinitialization based on dormant neurons (Sokar et al., 2023). These data-dependent regularizers regularizers control some property on data from the current task but may not control the property on data from a new task, which is needed to maintain plasticity.

Data-Independent Regularization, on the other hand, does not depend on any data. This category regularizes the parameters directly, which is particularly useful in continual learning when the data distribution is changing. Examples of data-independent regularization include L2 regularization towards zero, weight decay (for adaptive gradient methods), and regenerative regularization (Kumar et al., 2023). Our proposed approach of spectral regularization is data-independent and, moreover, explicitly targets a trainability condition for continual learning. Thus, we expect that it is particularly effective on maintaining trainability in continual learning compared to other data-independent regularizers.

## A.7 Spectral Regularization of Other Layers

**Normalization Layers**  Other layers using per-unit scaling, such as normalization layers (Ba et al., 2016; Ioffe and Szegedy, 2015), are also spectrally regularized. For example, denote $\gamma \in \mathbb{R}^d$ as the trainable scaling parameters for LayerNorm (Ba et al., 2016), then the element-wise product of the weights can be written as a diagonal weight matrix, $f(z) = \gamma \circ z = \text{Diag}(\gamma)z$. The maximum singular value of a diagonal matrix is the maximum entry, $\sigma_{max}(\text{Diag}(\gamma)) = \max_i |\gamma_i|$. However, optimisation with a maximum is problematic because the max is not differentiable. Furthermore, differentiable surrogates, such as log-sum-exp $((\log(\sum_i \exp(|\gamma_i|)) - 1)^2)$, only give a loose upper bound on the maximum which is problematic because we want to regularize the maximum towards one. Thus, we regularize each weight towards 1.

**Convolutional Layers**  Similar to other work, we reshape the convolutional weight tensor (with kernel size ($k \times k$, $d_{in}$ filters and $d_{out}$ filters) into a matrix of size $d_{out} \times (k \cdot k \cdot d_{in})$ (Yoshida and Miyato, 2017). The spectral norm of this reshaped matrix provides an efficient upper bound on the spectral norm of the Toeplitz matrix defining the convolution (Tsuzuku et al., 2018, Corollary 1).

## B  Experiment Details

All of our experiments used Adam (Kingma and Ba, 2015) where the default step size of 0.001 was selected after an initial sweep over $[0.005, 0.001, 0.0005]$. For all of our results, we use 10 random seeds and provided a shaded region corresponding to the standard error of the mean. For experiments on tiny-ImageNet SVHN2, CIFAR10 and CIFAR100, we used 4 seeds to sweep over the regularization strengths of $[0.01, 0.001, 0.0001]$, and found that 0.0001 worked well on tiny-ImageNet, CIFAR10 and CIFAR100 for all regularizers, whereas 0.001 worked best on SVHN2 for all regularizers.

Datasets and non-stationarities:

- MNIST: 28x28 greyscale images, 10 classes. Only the first 12800 datapoints were used for training, with a batch size of 512 and [40, 80, 120] epochs, and a total of 50 tasks.

- Fashion MNIST: 28x28 greyscale images, 10 classes. Only the first 12800 images, with a batch size of 512 and [40, 80, 120] epochs per task, and a total of 50 tasks.

- EMNIST: 28x28 greyscale images, 50 classes. We use the balanced version of the dataset, using the first 100000 datapoints. For random label non-stationarity, we used a batch size of 500 and 100 epochs per task, with a total of 50 tasks. For both label flipping and pixel permutation non-stationarities, we used a batch size of 500 and 20 epochs per task, with a total of 200 tasks.

- SVHN2: 32x32 color images, 10 classes. The first 50000 images were used for training, 5000 images from the test set were used for validation and the rest were used for testing. The batch size used was 500, we found 250 to be unreliable for learning due to high variance. For random label non-stationarity, 20 epochs per task and 25 tasks was enough to lose trainability for an unregularized network. For pixel permutation non-stationarity, 10 epochs and 100 tasks was enough to lose trainability for an unregularized network.

- CIFAR10: 32x32 color images, 10 classes. All of the 50000 images were used for training, 1000 images from the test set were used for validation and the rest were used for testing. We used a batch size of 250 and found this to be effective. For random label non-stationarity, 20 epochs per task and 30 tasks was enough to lose trainability for an unregularized network. For pixel permutation non-stationarity, 10 epochs and 100 tasks was enough to lose trainability for an unregularized network.

- CIFAR100: 32x32 color images, 100 classes. All of the 50000 images were used for training, 1000 images from the test set were used for validation and the rest were used for testing. We used a batch size of 250 and found this to be effective. For random label non-stationarity, 20 epochs per task and 30 tasks was enough to lose trainability for an unregularized network. For pixel permutation non-stationarity, 10 epochs and 100 tasks was enough to lose trainability for an unregularized network.

- tiny-ImageNet: 64x64 color images, 200 classes. All of the 100000 images were used for training, 10000 images were used for validation, and 10000 images were used for testing according to the predetermined split. We used a batch size of 250 and found this to be effective. For random label non-stationarity, 20 epochs per task and 30 tasks was enough to lose trainability for an unregularized network. For pixel permutation non-stationarity, 20 epochs and 100 tasks was enough to lose trainability for an unregularized network.

- Reinforcement Learning: We evaluate spectral regularization in RL control tasks from DMC benchmark (Tassa et al., 2020), with SAC method (Haarnoja et al., 2018) in demanding replay ratio (RR) regime (D'Oro et al., 2022) with 16 gradient updates per every new environments step. We choose this setup because a high RR regime leads to significant primacy bias (Nikishin et al., 2022) defined as *a tendency to overfit initial experiences that damages the rest of the learning process*.

Neural Network Architectures:

- MNIST, EMNIST, and Fashion MNIST: 4-layer MLP with 256 neurons per layer and relu activations. For experiments that use it, LayerNorm was applied after the linear weight matrix and before the non-linearity.

- tiny-ImageNet, CIFAR10, CIFAR100, and SVHN2: We follow the architectures used by Lee et al. (2024) which include both an off-the-shelf ResNet-18 with batch norm, as well as a Vision Transformer with an embedding dimension of 192, patch size of 4x4, 3 attention heads, 12 layers, layer normalization and a dropout rate of 0.1.
- Reinforcement Learning: Recent studies (Ball et al., 2023; Nauman et al., 2024) have demonstrated that, in this setup, only resets with layer normalization (Ba et al., 2016) serve as an effective mitigation strategy. We compare the performance of the SAC agent with both spectral regularization and layer normalization to two baseline agents: SAC with only layer normalization, and SAC with layer normalization plus resets. We use spectral regularization with a coefficient $1e-4$ for both the actor and critic. For every method, we use a single critic and architecture size of 2 layers and 256 neurons per layer for both actor and critic. Using a random policy, we prefill a replay buffer with $10,000$ transitions before starting the training. Replay buffer maximum size is 1 million transitions.

Metrics reported

- Figure 1: test accuracy at the end of training on a task
- Figure 2: test accuracy at the end of training on a task
- Figure 3: top: train accuracy at the end of training on a task, middle-top: average of the maximum singular value at each laeyr, middle-bottom: average distance between representation at the beginning of a task and at the end of a task, bottom: distance between current parameters and parameters at initialization.
- Figure 4: Area under curve for training accuracy as a function of regularization strength.
- Figure 5: Left: Return achieved by policy under evaluation, middle: area under curve of return as a function of the hyperparameter value, right-top: difference between estimated Q-value and Monte-carlo return under the current policy, right-middle: average gradient norm for the policy network, right-bottom: average parameter norm for the policy network.

# C ADDITIONAL EXPERIMENTS

## C.1 ABLATING HYPERPARAMETER $k$ FOR SPECTRAL REGULARIZATION

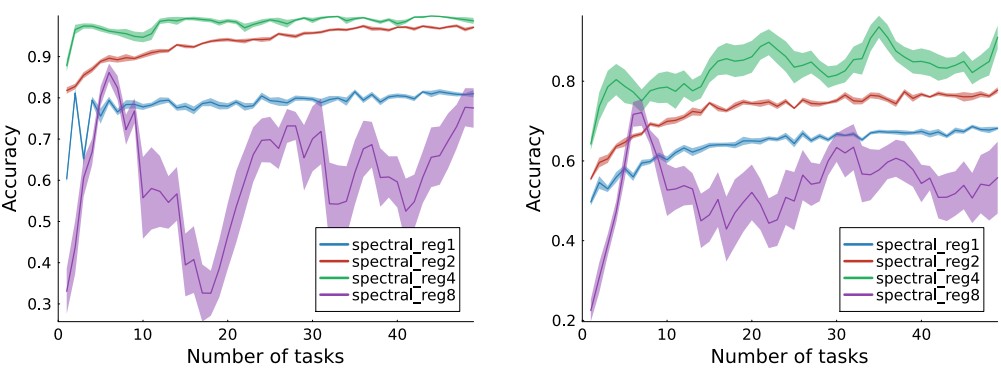

Figure 7: **Evaluating the choice of $k$ for spectral regularization for $k = 1, 2, 4, 8$ on MNIST (Right) and Fashion MNIST (left).** We found that $k = 2$ balance stability with effectiveness, and use this value throughout our experiments.

## C.2 RESULTS ON FASHION MNIST

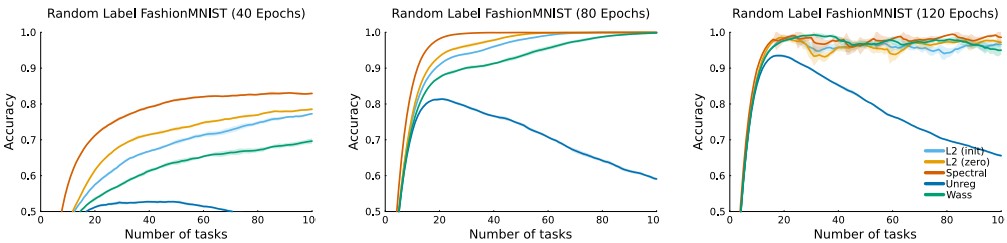

Figure 8: **Loss of Plasticity in Fashion MNIST.** Although the all the newtorks in this experiment use LayerNorm, loss of plasticity occurs without regularization: the performance decreases as a function of tasks, even with an increasing number of iterations.

## C.3 VISION TRANSFORMER CANNOT MEMORIZE RANDOM LABELS

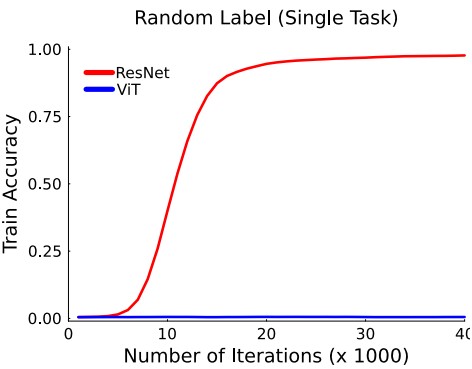

Figure 9: **ResNet and Vision Transformer on the random label memorization task using tiny-ImageNet.** The Vision Transformer architecture is particularly well-suited to tasks with structure from which generalization is possible. However, we found that its trainability on random labels is lacking. We were unable to get the Vision Transformer to memorize random labels even on a single task in ImageNet.

## C.4 ADDITIONAL SENSITIVITY RESULTS

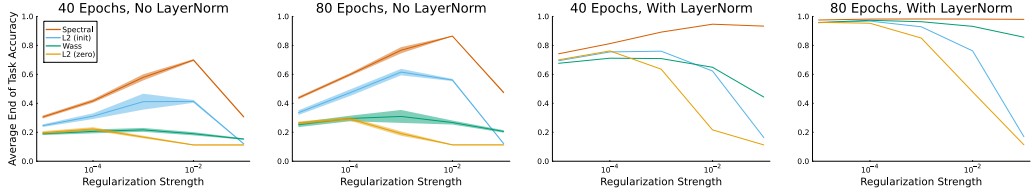

Figure 10: **Sensitivity analysis on regularization strength** Compared to other regularizers, spectral regularization is insensitive to regularization strength while sustaining higher trainability for any given regularization strength.

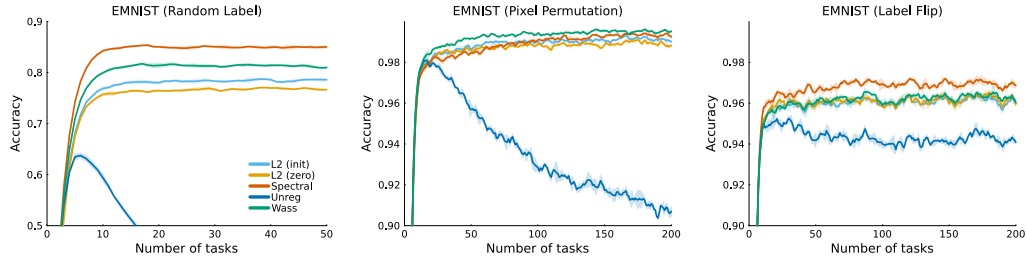

Figure 11: Sensitivity to type of non-stationarity on EMNIST. Spectral regularization is able to consistently maintain high trainability across random label assignment, pixel permutations, and label flipping. When applying random label assignment, L2 (init) and L2 (zero) are unable to attain high trainability as compared to spectral regularization.

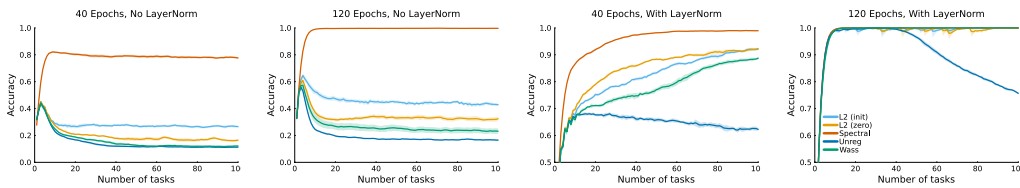

Figure 12: Sensitivity to number of epochs per task on MNIST with random label assignment. Without Layer Norm, spectral regularization is able to maintain trainability over other regularizers even when the number of epochs is low. Spectral regularization also synergizes with Layer Norm, improving initial performance on the first task, and improving over tasks.

**Robustness to Non-stationarity in Fashion MNIST**   Next, we explore whether our proposed regularizers are effective across different types of non-stationarity. To this end, we use EMNIST because the number of classes is large enough that label flipping induces loss of trainability (Elsayed and Mahmood, 2024). In Figure 11, we found that spectral regularization best maintained trainability across different non-stationarities.

**Varying the Number of Epochs Per Task**   Given that the continual learning problem outlined in Section 2 depends on the number of iterations, it may be the case that a higher number of epochs per task can mitigate loss of trainability. In Figure 12, we found that the number of epochs only delays loss of trainability (see Appendix C.2 for results on Fashion MNIST). Even when the number of epochs per task is high enough to reach 100% accuracy on the first task, loss of trainability eventually occurred without regularization. In contrast, when using spectral regularization, loss of trainabililty was consistently mitigated. We additionally found that spectral regularization is particularly effective at maintaining trainability when the number of epochs per tasks is low. The evidence for this is most striking without Layer Norm, in the top row, where spectral regularization was the only method capable of maintaining its trainability. Even though Layer Norm controls the spectral norm through the activations (Kim et al., 2021), adding Layer Norm alone was not enough to mitigate loss of trainability. However, Layer Norm is synergistic with various forms of regularization, with accuracy being improved regardless of the number of epochs per task.

## C.5 EFFECT OF REGULARIZATION ON CAPACITY IN A SINGLE TASK

The regularization strength for preventing loss of trainability must be sufficiently high, and this regularization strength may limit the capacity of the neural network. We now show the extended training of each regularizer using the same regularization strength used to prevent loss of trainability in Figure 13. Spectral regularization was not only best in preventing loss of trainability, it also achieved the highest accuracy when training to convergence on a single task. This means that spectral regularization constrains the capacity of the neural network the least, while still preventing loss of trainability.

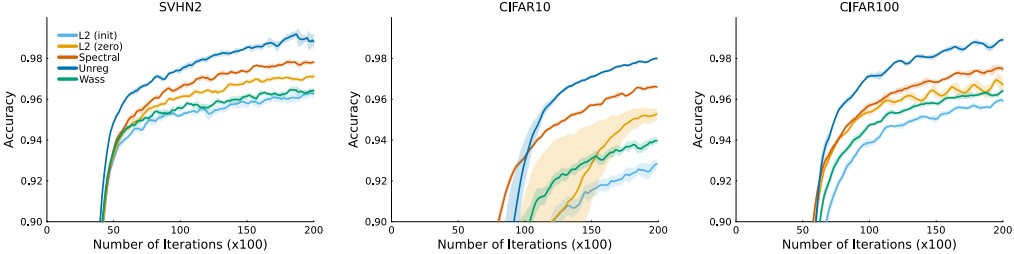

Figure 13: **Single task performance with ResNet-18.** Spectral regularization is least restrictive of the neural network capacity, evidenced by its ability to better fit the randomly assigned labels compared to other regularizers. The unregularized baseline is able to use its capacity fully, but at the cost of reduced trainability in later tasks.

## C.6 INVESTIGATING EFFECTS OF REGULARIZATION ON NEURAL NETWORK PROPERTIES

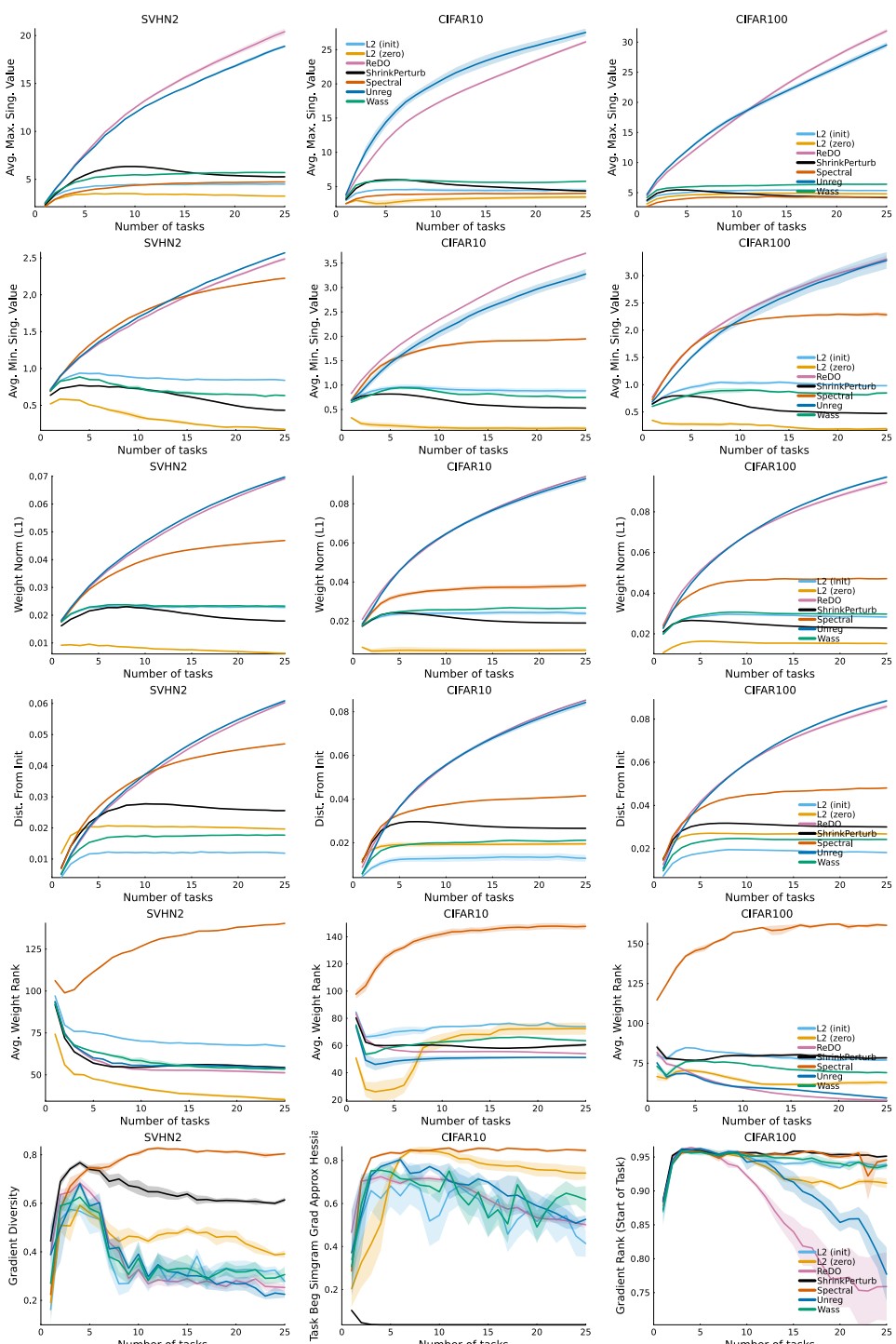

Figure 14: **Singular values, weight norms, stable ranks and effective gradient diversity**: Left: SVHN2, Middle: CIFAR10, Right: CIFAR100.

## C.7 INVESTIGATING GENERALIZATION ON CONTINUAL IMAGENET

We investigate the generalization performance of spectral regularization on Continual ImageNet. In Continual ImageNet, each task is to distinguish between two ImageNet classes. We use the network architecture and training protocol used in Kumar et al. (2023). We find that all regularization approaches tested achieve high generalization performance across tasks. The methods we compare are spectral regularization (SpectralRegAgent), L2 towards zero (L2Agent), L2 towards initialization (L2InitAgent), recycling dormant neurons (ReDOAgent), and no regularization (BaseAgent).

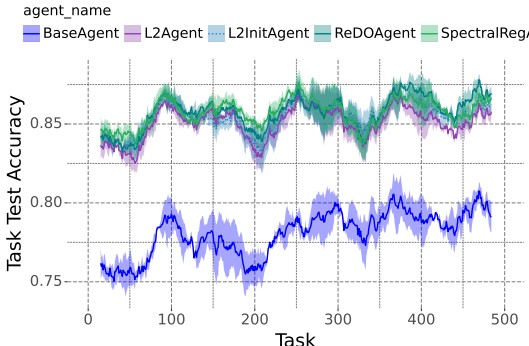

Figure 15: Generalization performance on Continual ImageNet.

## C.8 RESULTS ON INDIVIDUAL DMC ENVIRONMENTS

We report mean returns for DMC environments in Figure 16.

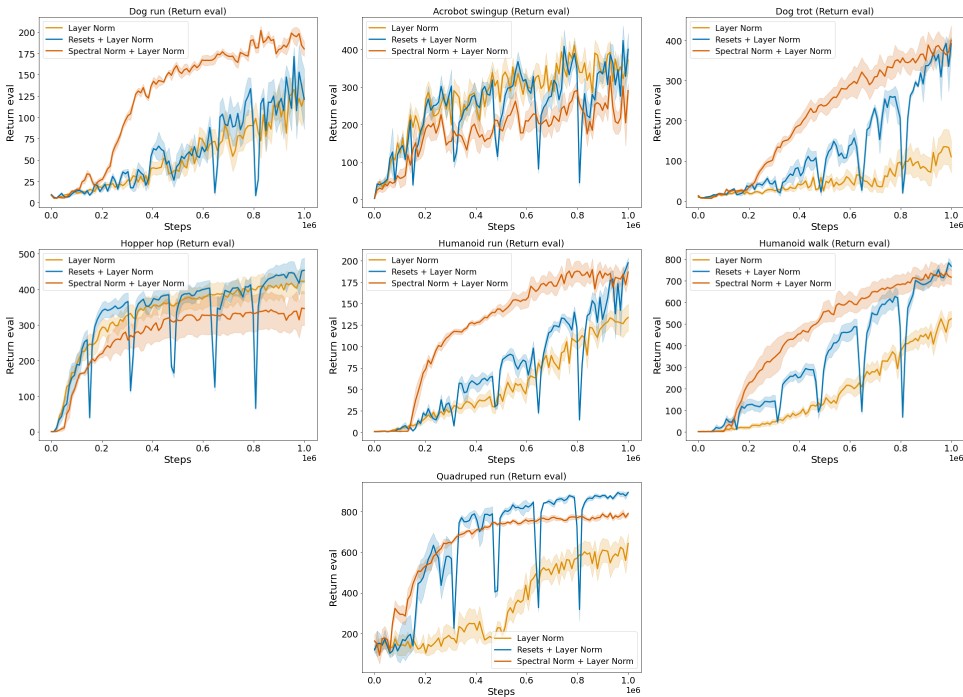

Figure 16: Mean and standard error of return for 7 DMC environments.

## C.9    COMPARING CONTINUAL BACKPROP, REDO AND SPECTRAL REGULARIZATION

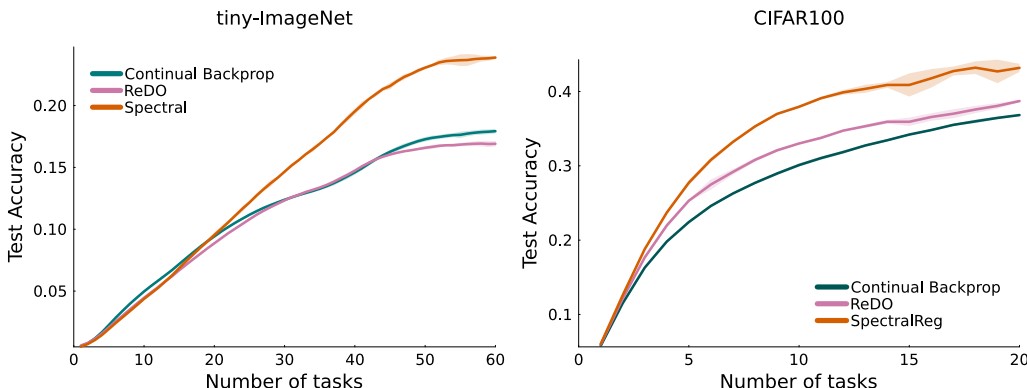

Figure 17: **Class-Incremental Learning on tiny-ImageNet and CIFAR100 with continual backprop.** For continual backprop, we used the suggested hyperparameters: a maturity threshold of 1000 and a replacement rate of $10^{-5}$. Continual backprop performs similarly to Recycling DOrmant neurons (ReDO). However, spectral regularization outperforms both methods.

## C.10    TABLE SUMMARY OF MAIN RESULTS

|               | Class Incremental | Label Flip      | Pixel Perm      |
|---------------|-------------------|-----------------|-----------------|
| L2 (init)     | 0.2313 (0.0013)   | 0.2571 (0.0013) | 0.0926 (0.0018) |
| L2 (zero)     | 0.2197 (0.0021)   | 0.2670 (0.0029) | 0.0873 (0.0015) |
| ReDO          | 0.1726 (0.0008)   | 0.2478 (0.0103) | 0.0754 (0.0025) |
| S-EWC         | 0.1615 (0.0016)   | 0.2655 (0.0011) | 0.0782 (0.0018) |
| ShrinkPerturb | 0.1778 (0.0053)   | 0.2451 (0.0051) | 0.1085 (0.0025) |
| Unreg         | 0.1749 (0.0007)   | 0.2307 (0.0056) | 0.0788 (0.0009) |
| Wass          | 0.2260 (0.0022)   | 0.2726 (0.0040) | 0.1057 (0.0019) |
| Spectral      | 0.2447 (0.0016)   | 0.2594 (0.0016) | 0.1104 (0.0029) |

Table 1: Table summarizing the results in the top row of Figure 1 (mean of final test accuracy and standard deviation, ResNet on tiny-ImageNet).

|               | Class Incremental | Label Flip      | Pixel Perm      |
|---------------|-------------------|-----------------|-----------------|
| L2 (init)     | 0.1890 (0.0042)   | 0.2822 (0.0045) | 0.1370 (0.0022) |
| L2 (zero)     | 0.1457 (0.0001)   | 0.2126 (0.0047) | 0.1188 (0.0012) |
| S-EWC         | 0.1185 (0.0052)   | 0.2230 (0.0079) | 0.0923 (0.0073) |
| ShrinkPerturb | 0.1171 (0.0011)   | 0.1392 (0.0068) | 0.1088 (0.0033) |
| Unreg         | 0.1226 (0.0074)   | 0.1936 (0.0029) | 0.0813 (0.0021) |
| Wass          | 0.1727 (0.0252)   | 0.2184 (0.0022) | 0.0800 (0.0010) |
| Spectral      | 0.2452 (0.0014)   | 0.2743 (0.0081) | 0.1392 (0.0006) |

Table 2: Table summarizing the results in the bottom row of Figure 1 (mean of final test accuracy and standard deviation, Vision Transformer on tiny-ImageNet).

|  | SVHN2 | CIFAR10 | CIFAR100 |
|---|---|---|---|
| L2 (init) | 0.7341 (0.0032) | 0.4830 (0.0038) | 0.2594 (0.0037) |
| L2 (zero) | 0.7409 (0.0027) | 0.4770 (0.0035) | 0.2563 (0.0023) |
| ReDO | 0.7356 (0.0018) | 0.4531 (0.0016) | 0.2485 (0.0012) |
| S-EWC | 0.7288 (0.0027) | 0.4612 (0.0018) | 0.2481 (0.0017) |
| CReLU | 0.7539 (0.0016) | 0.5002 (0.0028) | 0.2443 (0.0013) |
| ShrinkPerturb | 0.7548 (0.0015) | 0.5032 (0.0016) | 0.2546 (0.0008) |
| Unreg | 0.7277 (0.0036) | 0.4490 (0.0023) | 0.2444 (0.0018) |
| Wass | 0.7337 (0.0032) | 0.4849 (0.0052) | 0.2654 (0.0023) |
| Spectral | 0.7489 (0.0026) | 0.5082 (0.0052) | 0.2785 (0.0043) |

Table 3: Table summarizing the results in the top row of Figure 2 (mean of final test accuracy and standard deviation, ResNet on SVHN2, CIFAR10 and CIFAR100 with pixel permutation nonstationarity).

|  | SVHN2 | CIFAR10 | CIFAR100 |
|---|---|---|---|
| L2 (init) | 0.8163 (0.0028) | 0.5160 (0.0034) | 0.2267 (0.0023) |
| L2 (zero) | 0.6476 (0.1494) | 0.5251 (0.0019) | 0.2457 (0.0031) |
| S-EWC | 0.8003 (0.0040) | 0.4735 (0.0026) | 0.2371 (0.0030) |
| ShrinkPerturb | 0.3565 (0.1344) | 0.5357 (0.0062) | 0.2564 (0.0019) |
| Unreg | 0.7870 (0.0044) | 0.4610 (0.0041) | 0.2004 (0.0120) |
| Wass | 0.8098 (0.0035) | 0.4811 (0.0078) | 0.2132 (0.0027) |
| Spectral | 0.8248 (0.0016) | 0.5310 (0.0044) | 0.2775 (0.0024) |

Table 4: Table summarizing the results in the bottom row of Figure 2 (mean of final test accuracy and standard deviation, Vision Transformer on SVHN2, CIFAR10 and CIFAR100 with pixel permutation nonstationarity).

|  | tiny ImageNet | CIFAR10 | CIFAR100 |
|---|---|---|---|
| L2 (init) | 0.8206 (0.0172) | 0.8317 (0.0094) | 0.6499 (0.0287) |
| L2 (zero) | 0.7974 (0.0102) | 0.6536 (0.0143) | 0.5973 (0.0244) |
| ReDO | 0.2637 (0.0107) | 0.7387 (0.0561) | 0.2800 (0.0578) |
| S-EWC | 0.3607 (0.0934) | 0.7195 (0.0725) | 0.1980 (0.0287) |
| CReLU | 0.3587 (0.0086) | 0.7156 (0.0335) | 0.4722 (0.0121) |
| ShrinkPerturb | 0.5832 (0.0040) | 0.7451 (0.0064) | 0.6483 (0.0145) |
| Unreg | 0.2361 (0.0302) | 0.7371 (0.0297) | 0.2671 (0.0366) |
| Spectral | 0.8542 (0.0040) | 0.8159 (0.0072) | 0.7771 (0.0128) |

Table 5: Table summarizing the results in the bottom row of Figure 2 (mean of final train accuracy and standard deviation, ResNet on tiny ImageNet, CIFAR10 and CIFAR100 with random label nonstationarity).

