# OpenReview forum: "Learning Continually by Spectral Regularization"
_ICLR.cc/2025/Conference — ICLR 2025 Poster_

### Official Review · Reviewer_f6F3 · 2024-10-31

**Soundness:** 2
**Presentation:** 2
**Contribution:** 2
**Rating:** 6
**Confidence:** 3

**Summary:**

The paper introduces a new regularization to encourage neural networks to combat loss of plasticity phenomenon in a continual learning setting. The new approach is based on the observation that the maximum singular value of neural network layers grow over the course of training and avoiding this growth can help improve trainability of models in a continual learning setting. The proposed regularizer attempts to steer the maximum singular value of a layer to 1 (and 0 for the bias terms). The paper uses accepted datasets and tasks in continual learning literature to study the effectiveness of the proposed method over several baselines.

**Strengths:**

- The paper solves the problem of loss of plasticity in continual learning by addressing trainability.

- Section 3 and Section 4 in the paper are nicely written and is valuable to a general audience. Also, the paper provides at least two illustrative examples as well as some analysis to make the observation that the top singular value growth may cause training issues.

- The development of proposed solution via spectral regularization  is intuitive

**Weaknesses:**

* The proposed method to keep top singular value small bears some similarities to the following paper:
  - https://arxiv.org/abs/2303.06296

* The experimental setup used in the paper is incomplete and not very convincing in order to judge the empirical results:
  - The models used are on the smaller side (ResNet-18 and ViT)
  - ViT model size is not specified in the paper
  - The dataset used in Tiny-Imagenet which is not fully described. I believe the inputs are 64x64 based on my search but this detail is missing in the paper
While I am not an expert in continual learning area, there are recent papers that suggest that larger datasets (resolution and size) are available and used in continual learning studies

* The presentation of the experimental results and discussion section could be improved:
  - Figure 1 has legend only on the top left figure which makes it hard for the reader to follow how each method works across different continual learning scenarios
  - Figure 2 has a similar issue with legend
  - he paper does not define all the quantities plotted in Figure 3.
  - Since the paper focuses on trainability, additional graphs that show how trainability improves in CL setting might be useful in the presentation. Figure 3 provides a hint but in this case the shrink and perturb baseline shows better trainability as defined by singular value.

I'm unable to support acceptance at this point due questions on setup used in empirical analysis. The paper could benefit from an update that includes missing information that can give the reader an idea on exact metrics tracked, setup used, dataset details and other information.

**Questions:**

In addition to the points noted in weakness section, I wonder if the following statement is too strong
```
Note that these datasets and architectures encompass all continual supervised learning experimental settings considered in the loss of plasticity literature.
```

---

> ### Author Response · Authors · 2024-11-19
>
> We are pleased that the reviewer found our treatment of trainability for continual learning thorough, well-written and intuitive. We have added the provided reference, several clarifying sentences given the points that you raised, and additional table summaries of our main results for additional clarity (changes and new additions are in blue). Below we address your comments,
>
> >The proposed method to keep top singular value small bears some similarities to the following paper: https://arxiv.org/abs/2303.06296
>
> Thank you for the reference, we have included it as an additional citation in our submission. The linked paper applies spectral normalization (rather than regularization) in training transformers, an approach first popularized in training GANs [1]. Previous work has already demonstrated that spectral normalization is not effective for sustaining plasticity [2]. We describe why in the last paragraph of Section 4: the gradient dynamics introduced by spectral normalization is data-dependent, which is problematic for changing data distributions. Spectral regularization is data-independent, the regularizer depends only on the parameters and not on the data.
>
>
> >The experimental setup used in the paper is incomplete and not very convincing in order to judge the empirical results: The models used are on the smaller side (ResNet-18 and ViT)
>
> About architectures: The size of these models is on-par with previously published works. For example, other recently published work only trains MLPs, including the last 2 layers of a pre-trained resnet [3], end-to-end resnet-18 [2,4], and vision transformer [2]. Our submission trains all of these architectures end-to-end, while using more large-scale datasets, more baselines and more nonstationarities.
>
> >ViT model size is not specified in the paper
>
> We used the same architecture as [5], which was just published at ICML. The architecture has an embedding dimension of 192, patch size of 4x4, 3 attention heads, 12 layers, layer normalization and a dropout rate of 0.1. We have added these details to Appendix B
>
> >”The dataset used in Tiny-Imagenet which is not fully described. I believe the inputs are 64x64 based on my search but this detail is missing in the paper While I am not an expert in continual learning area, there are recent papers that suggest that larger datasets (resolution and size) are available and used in continual learning studies”
>
> Thank you for bringing this up. We have added additional details regarding this dataset. The inputs are indeed 64x64, and there are 200 classes in total.
>
> About datasets: the loss of plasticity literature is different from other work in continual learning that focuses on catastrophic forgetting. In comparison, the loss of plasticity phenomenon involves much longer training processes (Figures 1 and 2 involve up to 100 tasks, which can amount to almost 3000 epochs / almost 1m iterations). These long training processes have been evaluated primarily using tiny-imagenet, CIFAR100 and CIFAR10. For example, CIFAR10 is used in [2], CIFAR10 and MNIST variants are used in [4,6], and more recently tiny-imagenet is used in [3,5].

---

> ### Author Response · Authors · 2024-11-19
> **Official Comment by Authors Pt. 2**
>
> >The presentation of the experimental results and discussion section could be improved: Figure 1 has legend only on the top left figure
>
> Presenting as many results as we have, given all the curves and baselines we consider, can be quite tricky; we then opted to have the legend show up in a single plot to avoid clutter. ote that the colors for different baselines are consistent across different figures. Nevertheless, we have now added Tables 1-5 in Appendix C.10 in the appendix for additional clarity.
>
> >The paper does not define all the quantities plotted in Figure 3.
>
> Thank you for bringing this up. We have added these details to the appendix (see, “Metrics Reported” in Appendix B).
>
> >Since the paper focuses on trainability, additional graphs that show how trainability improves in CL setting might be useful in the presentation.
>
> Our motivation is indeed focused on trainability from an optimization perspective, but this is ultimately for tractability purposes. Outside of continual learning, trainability of neural networks is much more well-understood compared to generalization. Thus, our goal was to extend this understanding to trainability of continual learning, while ultimately in-service to generalization performance which we demonstrate in Figures 1 and 2.
>
> >Figure 3 provides a hint but in this case the shrink and perturb baseline shows better trainability as defined by singular value.
>
> To clarify: trainability is not defined by the largest singular value. Rather, regularizing the largest singular value is sufficient to sustain trainability. Shrink and perturb uses L2 regularization which regularizes all the singular values, rather than just the largest singular value. But this extra regularization of the other singular values worsens hyperparameter sensitivity and can overly regularize the parameters. We added a sentence in Section 4 (line 279) which emphasizes this and provide more details in Appendix A.5.
>
> >In addition to the points noted in weakness section, I wonder if the following statement is too strong: ‘Note that these datasets and architectures encompass all continual supervised learning experimental settings considered in the loss of plasticity literature.’
>
> The reviewer is correct to point out that it is a strong statement. We were careful to qualify this statement, as it specifically concerns the continual supervised learning experiments in the loss of plasticity literature. See above, "about datasets:" and "about architectures:" for detailed discussion on this point.
>
>
> [1] Miyato et al. (2018). Spectral Normalization for Generative Adversarial Networks. ICLR.
>
> [2] Lyle et al. (2023). Understanding Plasticity in Neural Networks. ICML.
>
> [3] Elsayed et al. (2024). Addressing Loss of Plasticity and Catastrophic Forgetting in Continual Learning. ICLR.
>
> [4] Dohare et al. (2024). Loss of plasticity in deep continual learning. Nature (632).
>
> [5] Lee et al. (2024). Slow and Steady Wins the Race: Maintaining Plasticity with Hare and Tortoise Networks. ICML.
>
> [6] Galashov et al. (2024). Non-Stationary Learning of Neural Networks with Automatic Soft Parameter Reset. NeurIPS.

---

> ### Comment · Reviewer_f6F3 · 2024-11-23
>
> > The linked paper applies spectral normalization (rather than regularization) in training transformers, an approach first popularized in training GANs [1]. Previous work has already demonstrated that spectral normalization is not effective for sustaining plasticity [2]. We describe why in the last paragraph of Section 4: the gradient dynamics introduced by spectral normalization is data-dependent, which is problematic for changing data distributions. Spectral regularization is data-independent, the regularizer depends only on the parameters and not on the data.
>
> Thanks to the authors for clarifying this point and for updating Section 4 with clarifying details
>
> I appreciate the authors rebuttal on info about datasets and models (networks) as well as maximum singular value being sufficient for trainability.
>
> > The paper does not define all the quantities plotted in Figure 3.
> > Thank you for bringing this up. We have added these details to the appendix (see, “Metrics Reported” in Appendix B).
> The updated table (C.10) is very helpful. I find that that the numbers for CIFAR-10 are interesting in that the proposed method is always not the best. Can the authors comment on what might be happening here? This type of detail is what was hard for me to parse from Figure 1 that lead me to make my original comment.
>
> Overall, I believe the authors have clarified several details that I was confused about and also improved the paper by clarifying missing details. I will raise my score as I believe this work is good and interesting to continual supervised learning researchers in the ML community.

---

> > ### Author Response · Authors · 2024-11-25
> >
> > Dear Reviewer,
> >
> > Thank you for taking the time to read and respond to our rebuttal. We appreciate you updating your score to reflect this exchange. Please let us know if you have any other questions or concerns.

---

> > > ### Comment · Reviewer_f6F3 · 2024-12-02
> > >
> > > No additional questions or comments from my side. Just writing this to close the loop and once again commend the authors on their excellent rebuttal. I remain positive about this research contribution

---

### Official Review · Reviewer_uBBh · 2024-11-04

**Soundness:** 3
**Presentation:** 3
**Contribution:** 3
**Rating:** 8
**Confidence:** 4

**Summary:**

The paper addresses plasticity loss in continual learning by proposing a spectral regularization technique. The proposed regularization technique regularizes the network’s weights at each layer towards a spectral norm of 1. This method is motivated by a spectral analysis of (continual) learning and the observation that standard neural network weight initializations exhibit good trainability and generalization. The paper’s proposed method is validated on a series of continual learning benchmark problems where it is shown to be performant and robust to the choice of regularization strength.

**Strengths:**

- Section 1 Introduction and Section 2 Problem Setting are well written with clarity and detail, effectively introducing the problem, the spectral regularization algorithm, and some intuition behind its formulation.
- The spectral analysis of continual learning well motivates the algorithm.
- The experiments consider a wide swathe of existing continual learning algorithms, both ResNet-18 and Vision Transformer architectures, and a variety of datasets and non-stationarities. Over this set of experiments, spectral regularization attains competitive performance and is robust to its choice of hyperparameter.

**Weaknesses:**

- Although spectral regularization is well motivated and the paper includes positive empirical results, one way to improve the paper would be to derive some theory illustrating the benefit of spectral regularization, and drawbacks of other regularization schemes, in continual learning.

**Questions:**

- Is there any particular reason why Continual Backprop was not evaluated in the experiments?

---

> ### Author Response · Authors · 2024-11-19
>
> We are pleased that the reviewer found our submission clearly written, well-motivated and with “wide swathe of existing continual learning algorithms, both ResNet-18 and Vision Transformer architectures, and a variety of datasets and non-stationarities.” We have added additional experiments comparing continual backprop, ReDO and spectral regularization in Appendix C.9 (changes and new content is in blue). Below we address your comments,
>
> >one way to improve the paper would be to derive some theory illustrating the benefit of spectral regularization, and drawbacks of other regularization schemes, in continual learning.
>
> Thanks, this is an important point and we have added a few clarifying sentences to the end of Section 4.
>
> Note that maintaining provably good trainability and generalization requires understanding neural network training dynamics. This is an active area of research, even in simpler non-continual learning settings.
>
> However, we discuss the relationship between L2 regularization and spectral regularization in Appendix A.5. Spectral regularization is a selective regularizer, which only regularizes the maximum singular value. In comparison, L2 regularization minimizes the sum of the singular values of the weights, which can lead to rank collapse if many singular values go to zero. L2 also constrains the parameter set more than spectral regularization, which only regularizes the largest singular value. We see evidence of this constraint in our experiments (Figure 3). L2 regularization prevents the parameters from deviating from the initialization compared to spectral regularization.
>
>
>
> >Is there any particular reason why Continual Backprop was not evaluated in the experiments?
>
>
> The dormant neuron phenomenon paper [1] demonstrates that their Recycling Dormant (ReDO) approach is competitive with or outperforms continual backprop. Both ReDO and continual backprop reset the weights of a neural network, but they different notions of utility. We thus selected ReDO as a representative reinitialization method. Nevertheless, we provide additional results in Appendix C.9 confirming that continual backprop and redo perform similarly, and that spectral regularization outperforms both of them.
>
> [1] Sokar et al. (2023). The Dormant Neuron Phenomenon in Deep Reinforcement Learning. ICML.
> Reviewer f6F3

---

> ### Author Response · Authors · 2024-11-25
> **Gentle reminder**
>
> Dear Reviewer,
>
> We would like to thank you for the time spent reviewing our submission.
>
> The discussion phase will be ending tomorrow, and we wanted to send this gentle reminder. We hope our earlier rebuttal answered the questions you raised, and that you appreciate our additional experiments including continual backprop. We would love to hear back from the reviewer. Please let us know if you have any other questions.

---

> > ### Comment · Reviewer_uBBh · 2024-11-30
> > **Response to Author(s)**
> >
> > I would like to thank the author(s) for their response to my questions. Nonetheless, I will be keeping my score of 8 as I found the original submission sufficient to deem a score of 8 and it still remains so.

---

### Official Review · Reviewer_YuMr · 2024-11-05

**Soundness:** 2
**Presentation:** 3
**Contribution:** 2
**Rating:** 5
**Confidence:** 4

**Summary:**

The paper considers the continual learning problem. In order to keep the stability of problem, the spectral regularizer is added to the objective function. Experimental results demonstrate the effective of the approach method.

**Strengths:**

This paper provide the spectral regularize to the objective function for improving the stability of the problem.

**Weaknesses:**

However, the added spectral regularize would bring in the cost of computation for the continual learning if facing large model and large data.

**Questions:**

1 through the initialization is the key to the trainability, singular values of Jacobian and the condition numbers would induce the new selected parameters, e.g., rank
2 how to effectively deal with computation of the spectral norm, i.e. spectral regularization,
3 the performance of the spectral is not always better than that of vision transformer and resents
4 what’s more, the datasets are not enough, please refer to the latest approach,
5 for the figure 5, what’s the phenomenon of suddenly decrease for the y-axis.

---

> ### Author Response · Authors · 2024-11-19
>
> We appreciate that the reviewer recognizes the effectiveness of our proposed spectral regularizer.  We have edited the submission to include additional clarification regarding the computational cost of our approach (changes are indicated in blue). Below we address your comments,
> >spectral regularize would bring in the cost of computation for the continual learning if facing large model and large data. /  how to effectively deal with computation of the spectral norm, i.e. spectral regularization
>
> Thanks for bringing up this point. We use power iteration for the spectral regularizer, which requires only matrix-vector products and scales computationally linearly in the number of parameters (similar to gradient descent). In practice, this amounts to only a small overhead compared to the baseline. On a 1080 TI, training with spectral regularization is approximately 14% slower than training without any regularization. We have added a clarifying sentence and footnote in Section 4 for this detail (line 270).
> >through the initialization is the key to the trainability, singular values of Jacobian and the condition numbers would induce the new selected parameters, e.g., rank
>
> Could the reviewer clarify the question being asked here?
>
> >the performance of the spectral is not always better than that of vision transformer and resents.
>
> Actually, spectral regularization is always better than the baseline unregularized vision transformer and resnet. This unregularized baseline is the worst performer in every experiment, whereas spectral regularization is always among the best methods. In several experiments, it is a substantial improvement (see Figure 1 (left), Figure 2 (right), and Figure 3). Can the reviewer point out what results they are referring to in which spectral regularization does not outperform the vision transformer or resnets?
> >what’s more, the datasets are not enough, please refer to the latest approach
>
> Could the reviewer reviewer what they mean by the latest approach? The scope of our experiments, both in the datasets and the architectures, is on par with recently published works.  For example, other recently published work only trains MLPs, including the last 2 layers of a pre-trained resnet [2], end-to-end resnet-18 [1,3], and vision transformer [1]. Our submission trains all of these architectures end-to-end while using more large-scale datasets, more baselines and more types of nonstationarity.
>
> [1] Lyle et al. (2023). Understanding Plasticity in Neural Networks. ICML.
>
> [2] Elsayed et al. (2024). Addressing Loss of Plasticity and Catastrophic Forgetting in Continual Learning. ICLR.
>
> [3] Dohare et al. (2024). Loss of plasticity in deep continual learning. Nature (632).
>
> >for the figure 5, what’s the phenomenon of suddenly decrease for the y-axis.
>
> In that set of experiments, we used the a single hyperparameter for the regularization of both the policy and the value networks which can cause the “M” shape sensitivity curve. In supervised learning with a single network and objective, there is usually a simple quadratic-like shape in the sensitivity (see Figure 4). In reinforcement learning we  train two networks, a value network and a policy network. The policy and value have different objectives and architectures, which mean their sensitivity curves will be differently shaped quadratics. When the two quadratics are added together, we get the “M” shape we see in Figure 5.

---

> ### Author Response · Authors · 2024-11-25
> **Gentle reminder**
>
> Dear Reviewer,
>
> We would like to thank you for the time spent reviewing our submission.
>
> The discussion phase will be ending tomorrow, and we wanted to send this gentle reminder. We have done our best to answer the comments you raised, as well as incorporate your suggestions. We would love to hear back from the reviewer and whether we have addressed their concerns.

---

> ### Author Response · Authors · 2024-12-02
>
> Dear Reviewer,
>
> The extended discussion phase will be ending today, and we wanted to send this gentle reminder. We have done our best to answer the comments you raised, as well as incorporate your suggestions. We would love to hear back from the reviewer and whether we have addressed their concerns.

---

### Official Review · Reviewer_veM9 · 2024-11-06

**Soundness:** 3
**Presentation:** 3
**Contribution:** 2
**Rating:** 6
**Confidence:** 4

**Summary:**

The authors derive a new spectral regularizer for continual learning inspired by the observation that the singular values of the neural network parameters at initialization are an important factor for trainability during early phases of learning. Under such regularization they keep the maximum singular value of each layer close to one.

Under such spectral regularization they claim that they can directly ensure gradient diversity throughout training, which promotes continual trainability, while minimally interfering with performance in a single task.

To validate the utility of the proposed regularization and the claims they present empirical evaluation. The experimental results shows how the proposed spectral regularizer can sustain trainability and performance across a range of model architectures in continual supervised and reinforcement learning settings. They try to demonstrate better training in individual tasks, trainability as new tasks arrive, and better generalization performance.

**Strengths:**

- The continual learning topic is important area and loss of plasticity is fundamental challenge.
 - The approach is interesting and the idea to regularize the maximum singular value of each layer close to one seems novel.
 - The presentation of the work is nice, while the motivation is appropriate.
 - The authors present results for several experiments, evaluating different properties.

**Weaknesses:**

- It appears like a marginal improvement and novelty with respect to L2 regularization.
- It seems that there is a connection to L2 regularization, or L2 regularization towards initialization of parameters which is a sort of spectral regularization.
- Deeper understanding/analysis of the proposed and the one above seems important and might be beneficial, but I was not able to find it in the paper L2 norm vs spectral norm in this context.
- In the experimental section, looking at the graph and the results it seems that the improvements are marginal, or not highlighted significantly.
- In the same experimental section, I was able to see only graphs and plots, was not able to see actual table or values, to be able to apricate the achieved results more.

**Questions:**

How different the propose approach is from L2 regularization on the same set of parameters?

---

> ### Author Response · Authors · 2024-11-19
>
> We are pleased that the reviewer appreciates the presentation of our paper and considers the ideas novel. We have edited the submission to include results in a table format in the appendix, as well as clarifying sentences in the main text (changes are indicated in blue). Below we address your comments,
>
> >It appears like a marginal improvement and novelty with respect to L2 regularization.
>
> Empirically, spectral regularization is a large improvement over L2 regularization in several of our experiments, e.g. Figure 1 (left), Figure 2 (right), and Figure 3. Moreover, spectral regularization is more robust to its hyperparameter and always among the 1 or 2 best-performing methods in all of our experiments.
>
> In terms of novelty, the reviewer correctly notices that “the idea to regularize the maximum singular value of each layer close to one seems novel.” Indeed, regularizing the spectral norm of the weight matrices has been explored in other contexts. For example, it has been used to improve robustness to input perturbations [1]. However, its use in continual learning and to fight loss of plasticity, as well as the exact way of regularizing the norm is new. Unlike the usual norm-based regularization methods, we do not intend to make the norm as small as possible (with a minimum penalty achieved with a zero weight matrix) but regularize the largest singular values to be close to 1. This is also a clear distinction compared to L2 regularization, with multiple benefits: (i) we achieve better control of the condition number, and (ii) the regularization is much milder than the weight matrices are free to change in any direction that is not affecting the largest singular value, avoiding unnecessary restrictions for quickly fitting the data, as described in more detail below.
>
> [1] Yoshida and Miyato. (2017). Spectral Norm Regularization for Improving the Generalizability of Deep Learning. ArXiv
>
>
> >It seems that there is a connection to L2 regularization, or L2 regularization towards initialization of parameters which is a sort of spectral regularization. Deeper understanding/analysis of the proposed and the one above seems important and might be beneficial, but I was not able to find it in the paper L2 norm vs spectral norm in this context.
>
> Thanks for pointing this detail out, which we will emphasize in the main paper. We have added a few sentences to the last paragraph of Section 4 to make this more clear. Note that in Appendix A.5, we actually describe the differences between L2 regularization and spectral regularization. L2 regularization minimizes the sum of the singular values of the weights. This can lead to rank collapse if many singular values go to zero. This also constrains the parameter set more than spectral regularization, which only regularizes the largest singular value. We see evidence of this constraint in our experiments. L2 regularization prevents the parameters from deviating from the initialization compared to spectral regularization. L2 regularization towards initialization can avoid rank collapse because it minimizes the sum of the singular values of the weights minus the sum of the singular values of the initial weights, but again, it provides much less freedom for actually fitting the data than our spectral regularizer.
>
>
>
> > In the experimental section, looking at the graph and the results it seems that the improvements are marginal, or not highlighted significantly.
>
> Notice that spectral regularization is always amongst the best-performing methods in all experiments. Moreover, in several experiments, spectral regularization was significantly better than any other baseline: Figure 1 (left), Figure 2 (right), Figure 3. Another benefit that we highlight in our experiments is that spectral regularization is also highly robust to the architecture, dataset, nonstationarity and hyperparameter.
>
> > In the same experimental section, I was able to see only graphs and plots, was not able to see actual table or values, to be able to appreciate the achieved results more.
>
> The graphical presentation we use is better suited to showing continual learning performance, particularly when studying loss of plasticity as a function of the task. We have added Tables 1-5 in Appendix C.10 summarizing the results in Figures 1, 2, and 3. We have added a footnote in Section 5.2 highlighting this additional information.
>
> >How different the propose approach is from L2 regularization on the same set of parameters?
>
> If the reviewer is referring to an empirical comparison, then the L2 (zero) baseline included in our results is L2 regularization on the same architecture. We also describe the theoretical differences in Appendix A.5, and have added a few sentences to the end of Section 4 to make this explicit.

---

> ### Author Response · Authors · 2024-11-25
> **Gentle reminder**
>
> Dear Reviewer,
>
> We would like to thank you for the time spent reviewing our submission.
>
> The discussion phase will be ending tomorrow, and we wanted to send this gentle reminder. We have done our best to answer the comments you raised, as well as incorporate your suggestions. We would love to hear back from the reviewer and whether we have addressed their concerns.

---

> > ### Comment · Reviewer_veM9 · 2024-11-25
> >
> > Dear authors,
> >
> > Thanks for providing your feedback. I understand more the highlighted difference that I raised related to L2 norm and spectral regularisation.
> >
> > Unfortunately, I'm still not convinced by the efficiency, i.e., the empirical results. Therefore, I keep my score unchanged.

---

> > > ### Author Response · Authors · 2024-11-26
> > >
> > > We are not sure why the reviewer is unconvinced about the superior empirical performance of our results. Experimenting with all datasets and a set of representative architectures used in the continual learning literature for classification problems, we show that (i) our method is better than the best baseline in 10 out of 12 problems; (ii) our results also attest to the robustness of our method, given that it is always in the top 2 best-performing methods (the best baseline is in the top 2 only 7 times out of 12) -- see a table summarizing this below. Furthermore, we also show that our method is at least on par with the best baseline on a reinforcement learning problem, where the best baseline is only better than our method at the very end of the training for the original problem and our method becomes superior once the problem is varied a tiny bit (suggesting that the baseline might have been overfitted to the actual setting).
> > >
> > >
> > >
> > >
> > > |                 	| Spectral Regularization | L2 Regularization (towards 0) |
> > > |---------------------|:-----------------------:|:-----------------------------:|
> > > | CI-tinyImageNet 	|        	1        	|           	2           	|
> > > | LF-tinyImageNet 	|       	2*        	|          	2*           	|
> > > | PP-tinyImageNet 	|       	1*        	|           	3           	|
> > > | CI-tinyImageNet+vit |        	1        	|           	2           	|
> > > | LF-tinyImageNet+vit |       	1*        	|          	1*           	|
> > > | PP-tinyImageNet+vit |        	1        	|           	2           	|
> > > | PP-svhn2        	|       	2         	|          	4*           	|
> > > | PP-cifar10      	|       	1*        	|          	4*           	|
> > > | PP-cifar100     	|        	1        	|          	2*           	|
> > > | PP-svhn2+vit    	|        	1        	|           	2           	|
> > > | PP-cifar10+vit  	|       	1*        	|          	3           	|
> > > | PP-cifar100+vit 	|        	1        	|           	5           	|
> > >
> > >
> > > Note:
> > > * The table above lists the ranking, with 1 indicating the method achieves the highest test accuracy..
> > > * An asterisk (*) means that the method is tied at that position due to overlapping error bars. For example, if A and B are tied in first place, then both get a ranking of 1\* and C would be given a ranking of 2.
> > > * Spectral regularization is always equal to or better than L2 regularization, in all of our experiments.
> > > * Moreover, there are many instances of spectral regularization greatly out-performing L2 and other baselines, which can be readily seen from the tables and figures in our main paper.
> > >
> > > Legend:
> > > * CI = Class-Incremental
> > > * LF = Label-flip
> > > * PP = Pixel Permutation
> > >
> > >
> > > We would be genuinely curious to hear to what extent these results do not demonstrate the effectiveness of our method and what other results are missing to show this convincingly. Thanks!

---

### Meta-Review · Area_Chair_6Unn · 2024-12-21

**Metareview:**

The paper proposes spectral regularization scheme for continual learning tasks. Instead of simple L2 regularization, the proposed scheme aims to regularize the largest singular value of the weight matrices close to 1. While spectral regularization has been widely used for improving generalization capability of neural networks, directly applying it to continual learning seems to be new. A downside of it is its additional requirement of computation complexity to compute the regularization term. Empirical evaluation shows the proposed method is effective compared to other baselines for task-agnostic continual learning settings. Although the improvement is not a groundbreaking of a kind, the idea seems to be worth publishing in ICLR. So the decision is Accept.

Additionally, https://www.sciencedirect.com/science/article/pii/S0167865524001909 seems to be a relevant work, so it would be good to include it in the final version and address the difference between their method.

**Additional Comments On Reviewer Discussion:**

Reviewer uBBh and YuMr were not very informative as they either did not respond to author's rebuttal or responded with a very short comment without much information.

The other two reviewers, veM9 and f6F3, were engaged in the discussion and provided useful opinions for making the decision. veM9 clarified the difference between L2 regularization and the proposed method and also asked for quantitative numbers of the results. f6F3 clarified with various experimental setups.

---

### Decision · Program_Chairs · 2025-01-22

Accept (Poster)